# Coupling between the DEAD-box RNA helicases Ded1p and eIF4A

Zhaofeng Gao[1,2,3], Andrea A Putnam[1,2,3], Heath A Bowers[1,2,3], Ulf-Peter Guenther[1,2,3], Xuan Ye[1,2,3], Audrey Kindsfather[4], Angela K Hilliker[4], Eckhard Jankowsky[1,2,3]*

[1]Center for RNA Molecular Biology, Case Western Reserve University, Cleveland, United States; [2]Department of Biochemistry, Case Western Reserve University, Cleveland, United States; [3]School of Medicine, Case Western Reserve University, Cleveland, United States; [4]Department of Biology, University of Richmond, Richmond, United States

**Abstract** Eukaryotic translation initiation involves two conserved DEAD-box RNA helicases, eIF4A and Ded1p. Here we show that *S. cerevisiae* eIF4A and Ded1p directly interact with each other and simultaneously with the scaffolding protein eIF4G. We delineate a comprehensive thermodynamic framework for the interactions between Ded1p, eIF4A, eIF4G, RNA and ATP, which indicates that eIF4A, with and without eIF4G, acts as a modulator for activity and substrate preferences of Ded1p, which is the RNA remodeling unit in all complexes. Our results reveal and characterize an unexpected interdependence between the two RNA helicases and eIF4G, and suggest that Ded1p is an integral part of eIF4F, the complex comprising eIF4G, eIF4A, and eIF4E.

*For correspondence: exj13@ case.edu

**Competing interests:** The authors declare that no competing interests exist.

## Introduction

Translation initiation in eukaryotes involves at least 12 distinct protein factors, including two highly conserved DEAD-box RNA helicases, eIF4A and Ded1p (DDX3 in human) (*Aitken and Lorsch, 2012*; *Hinnebusch, 2014*; *Parsyan et al., 2011*). DEAD-box RNA helicases, whose name is inspired by a characteristic sequence motif (D-E-A-D, in single letter code), are enzymes that utilize ATP to bind and remodel RNA and RNA-protein complexes (*Linder and Jankowsky, 2011*). While *S. cerevisiae* eIF4A (~$10^5$ copies per cell) is roughly five times more abundant in the cell than Ded1p (*Firczuk et al., 2013*), both helicases are essential for viability (*Linder and Slonimski, 1989*; *Giaever et al., 2002*).

eIF4A unwinds RNA duplexes in vitro (*Merrick, 2015*). In the cell, the helicase interacts with the large scaffolding protein eIF4G that also binds the cap-binding protein eIF4E (*Hinnebusch, 2014*; *Parsyan et al., 2011*). The complex containing eIF4G, eIF4E and eIF4A is called eIF4F (*Merrick, 2015*). eIF4A binds to a MIF4G domain of eIF4G (*Schütz et al., 2008*), and this interaction stimulates RNA- and ATP-binding, and ATP-dependent RNA unwinding activity of eIF4A in vitro (*Hilbert et al., 2011*; *Rajagopal et al., 2012*). In translation initiation, eIF4A is thought to be critical for steps during and subsequent to the recruitment of the 40S ribosome subunit to the mRNA (*Parsyan et al., 2011*). In *S. cerevisiae*, eIF4A appears to promote at least one step necessary for the translation of most or all mRNAs (*Sen et al., 2015*).

Ded1p unwinds RNA duplexes in vitro as a trimer, whose formation requires the C-terminus (*Putnam et al., 2015*). Ded1p also facilitates strand annealing, RNA structure conversions, and the remodeling of RNA-protein complexes (*Bowers et al., 2006*; *Iost et al.,1999*; *Yang et al., 2007b*; *Yang and Jankowsky, 2005*). Both, Ded1p and human DDX3X interact with the cap binding protein eIF4E (*Senissar et al., 2014*; *Shih et al., 2008*), and with eIF4G (*Hilliker et al., 2011*; *Soto-*

*Rifo et al., 2012*). Ded1p uses its C-terminus to bind to the C terminus of eIF4G, a region distinct from the eIF4A binding site in eIF4G (*Hilliker et al., 2011*). The interaction with eIF4G interferes with Ded1p oligomerization and thus inhibits RNA unwinding by the helicase (*Putnam et al., 2015*). In translation initiation, Ded1p and DDX3X have been implicated in the mRNA recognition by eIF4F, recruitment of the 40S ribosome subunit to the mRNA, the scanning process, start codon selection and joining of the two ribosomal subunits (*Sharma and Jankowsky, 2014*). Misregulation of DDX3X expression and mutations in the helicase core of DDX3X are associated with tumors, including medulloblastoma, lung- and colorectal cancer and T-cell lymphoma (*Bol et al., 2015*; *Heerma van Voss et al., 2015*; *Jiang et al., 2015*; *Pugh et al., 2012*). DDX3 is also targeted by multiple pathogenic viruses including HIV, HCV, other flaviviridae, poxviridae, and HBV (*Valiente-Echeverría et al., 2015*).

Previous data indicate genetic interactions between eIF4A and Ded1p (*de la Cruz et al., 1997*; *Senissar et al., 2014*), and a recent, transcriptome-wide study in yeast shows overlapping, but not identical functions of eIF4A and Ded1p in translation initiation (*Sen et al., 2015*). While these data suggest a possible link between the two helicases, Ded1p and eIF4A have been generally viewed as separately functioning factors. Despite the central biological roles of the two enzymes, their level of conservation, and their implication in numerous diseases, it is unknown whether and how eIF4A and Ded1p are linked biochemically and physically, and if their functions are coordinated. The answer to these questions is of considerable importance for understanding eukaryotic translation initiation on the molecular level. However, experimentally addressing the link between the two enzymes presents significant challenges. First, as RNA helicases, eIF4A and Ded1p share basic biochemical properties including RNA binding, unwinding, ATP binding and ATP turnover. Therefore, enzymatic readouts are similar and potentially difficult deconvolute. Second, both helicases interact with eIF4G, which also binds RNA. The multitude of possible individual interactions significantly complicates the analyses of coupling between the proteins. Third, interactions between the proteins are likely transient or dynamic, given that both, Ded1p and eIF4A bind and hydrolyze ATP, which in turn modulates RNA binding by the helicases (*Linder and Jankowsky, 2011*). Functional analysis of dynamic multi-component assemblies is not well developed, even though such RNA-protein complexes are numerous in biology.

Here, we have confronted the experimental challenges associated with links between Ded1p, eIF4A, eIF4G, ATP and RNA. Using quantitative biochemical and enzymological approaches, we show that Ded1p and eIF4A directly interact with each other and that both helicases simultaneously interact with eIF4G. We devise an unbiased, comprehensive thermodynamic framework for the interactions between Ded1p, eIF4A, eIF4G, RNA and ATP. This framework indicates that eIF4A, with and without eIF4G, acts as a modulator for activity and substrate preferences of Ded1p, and that Ded1p functions as the RNA remodeling unit in complexes with eIF4A. Our observations reveal an unexpected interdependence between both RNA helicases and eIF4G, and suggest that Ded1p is an integral part of the eIF4F complex.

## Results

### Direct interactions between Ded1p, eIF4A and eIF4G

We identified eIF4A in a screen for genes that suppress a cold-sensitive growth defect conferred by *DED1* mutations (*ded1-tam*, *Hilliker et al., 2011*). Overexpression of eIF4A partially restored growth at non-permissive temperatures (*Figure 1A*), suggesting that eIF4A can promote Ded1p's function, but not completely substitute for its tasks. This observation is consistent with the notion that Ded1p and eIF4A have overlapping, but not identical functions in translation initiation (*Sen et al., 2015*).

To further examine the connection between Ded1p and eIF4A in the cell, we tested whether the eIF4A inhibitor hippuristanol exacerbated the temperature-sensitive growth defect associated with the *ded1-95* strain (*Burckin et al., 2005*). *Ded1-95* bears a T408I mutation, which impairs RNA binding by Ded1p in vitro (*Figure 1—figure supplement 1*). Hippuristanol inhibits the activities of eIF4A, but not those of Ded1p (*Bordeleau et al., 2006*). Here, Hippuristanol diminished growth of the *ded1-95* strain, compared to the WT *ded1* strain (*Figure 1B*), indicating that targeted inhibition of eIF4A can impact a growth defect associated with a mutation in *DED1*.

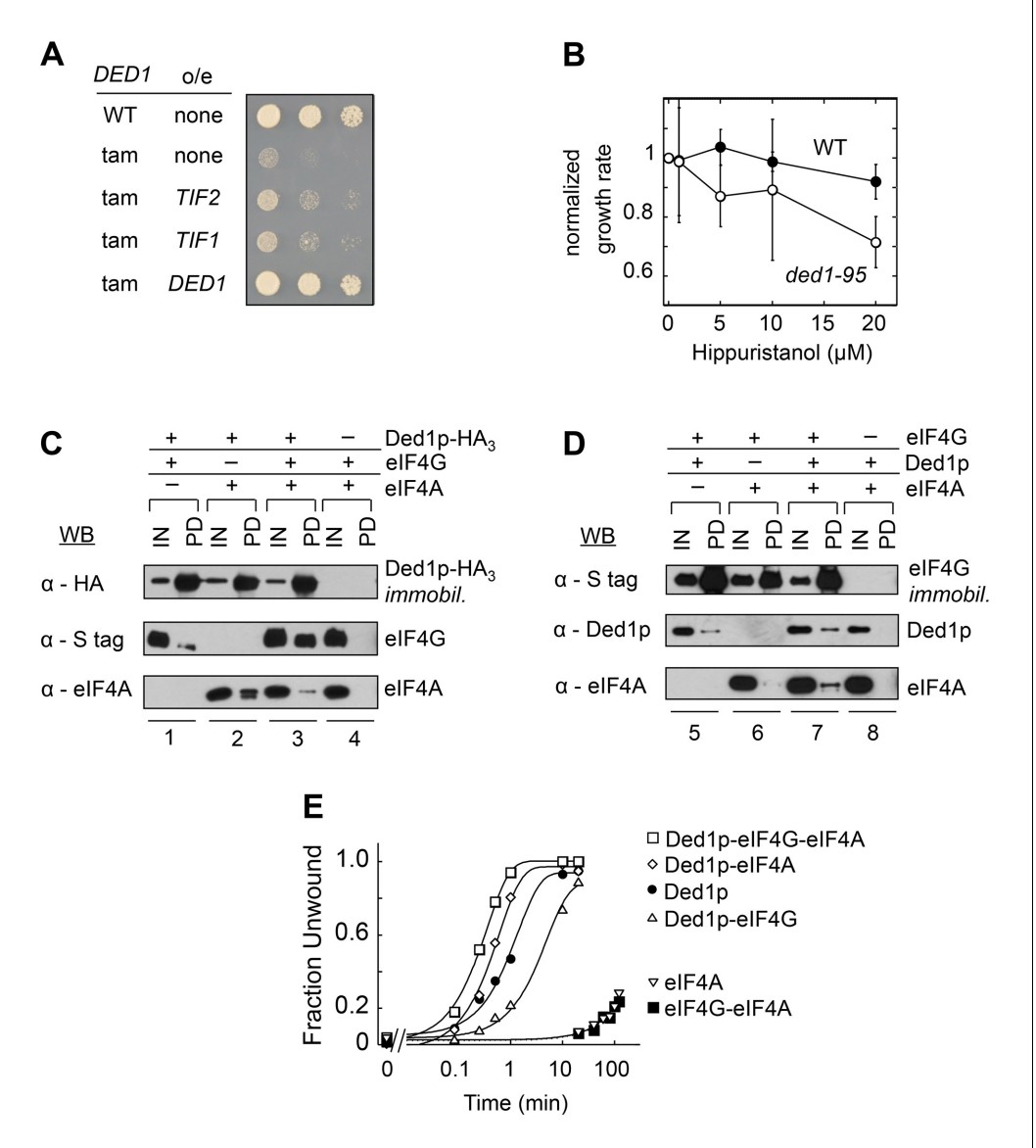

**Figure 1.** Direct interactions between Ded1p, eIF4A and eIF4G. (**A**) Effect of *EIF4A* (*TIF1*,*TIF2*) overexpression (o/e) on growth of the *DED1*-tam allele. Yeast with either wild type *DED1* (WT), or *DED1*-tam (tam) as a sole copy of *DED1* were transformed with a high copy plasmid containing either no insert (none) or *TIF1, TIF2*, or *DED1*. Serial dilutions of each strain were spotted at identical concentrations on selective media and grown at 16°C. (**B**) Impact of increasing concentrations of the eIF4A inhibitor hippuristanol on the growth of yeast with either wild type *DED1* (WT), or the temperature sensitive *ded1-95* (Ded1p$^{T408I}$, *Figure 1—figure supplement 1*) allele as the sole copy. Growth rates were measured at 37°C and normalized to the growth rate without inhibitor. Error bars represent one standard deviation of three independent measurements. (**C**) Pull-down of purified eIF4A, eIF4G, or both, with immobilized HA-tagged Ded1p. Reactions were performed in the presence of RNases. EIF4G contained an S-tag. Input (IN) lanes show 3% of total reaction volume. Pull-down (PD) lanes show entire sample. Relative band intensities (pull-down compared to input): reaction 1, Ded1p: 460% ± 18%, eIF4G: 31% ± 12%; reaction 2: Ded1p: 413% ± 58%, eIF4A: 96% ± 55%; reaction 3, Ded1p: 592% ± 113%, eIF4G: 74% ± 9%, eIF4A: 25% ± 21%. Errors represent one standard deviation of multiple independent experiments. (**D**) Pull-down of purified eIF4A, Ded1p, or both, with immobilized, S-tagged eIF4G. Reactions were performed in the presence of RNases. Input (IN) lanes show 3% of total reaction volume. Pull-down (PD) lanes show entire sample. Relative band intensities (pull-down compared to input): reaction 5, eIF4G: 440% ± 113%, Ded1p: 14% ± 7%; reaction 6: eIF4G: 280% ± 29%, eIF4A: 3% ± 2%; reaction 7, eIF4G: 473% ± 27%, Ded1p: 33% ± 11%, eIF4A: 15% ± 5%. Errors represent one standard deviation of multiple independent experiments. (**E**) Unwinding activity of Ded1p, eIF4A, with and without eIF4G, and in combination. Representative unwinding reaction with 0.5 nM RNA duplex (16 bp, 25 nt 3'-overhang, for

*Figure 1 continued on next page*

*Figure 1 continued*

sequences see *Supplementary file 1A*), 0.1 µM Ded1p, 0.1 µM eIF4G, 2 µM eIF4A, 4 mM ATP (0.2% v/v DMSO). Lines represent the best fit to the integrated first order rate law. See also *Figure 1—figure supplement 1* and *Supplementary file 1A*.

The following figure supplement is available for figure 1:

**Figure supplement 1.** Ded1p[T408I] is deficient in RNA binding.

Collectively, these observations expand the scope of genetic interactions between the two DEAD-box helicases. However, the data did not reveal the physical nature of the link between Ded1p and eIF4A.

To probe physical connections between the two helicases, we performed pull-down experiments with recombinant, purified proteins (*Figure 1C,D*). Previous data had shown that both helicases bind to distinct regions of the scaffolding protein eIF4G (*Hilliker et al., 2011*; *Merrick, 2015*). It was thus important to determine whether Ded1p could directly interact with eIF4A, and to probe whether both proteins could interact simultaneously with eIF4G. Reactions were performed with RNases to exclude possible RNA bridging of the three RNA binding proteins. EIF4A bound directly to Ded1p (*Figure 1C*, lanes 2). This observation indicates a direct interaction between the two DEAD-box helicases that does not depend on RNA or another factor. EIF4A increased the amount of eIF4G bound to immobilized Ded1p (*Figure 1C*, compare lanes 1 and 3). As expected, Ded1p bound to immobilized eIF4G (*Figure 1D*, compare lanes 5 and 7). Ded1p increased the amount of bound eIF4A to eIF4G (*Figure 1D*, compare lanes 6 and 7). Together, the data reveal a direct interaction between Ded1p and eIF4A and promotion of Ded1p-eIF4G binding by eIF4A. The observations show that interactions between Ded1p, eIF4G, and eIF4A are not mutually exclusive and suggest that all three proteins can interact simultaneously.

## Complexes between Ded1p, eIF4A and eIF4G are unwinding-competent

We next probed whether complexes formed between Ded1p and eIF4A alone or in combination with eIF4G were biochemically active. To this end, we measured ATP-dependent RNA unwinding activity for the various protein combinations using an RNA substrate with a 16 bp duplex. This substrate was unwound by Ded1p with a much larger apparent rate constant than seen for eIF4A (*Figure 1E*), consistent with expectations based on previous studies (*Rajagopal et al., 2012*; *Yang et al., 2007a*). Nevertheless, the addition of eIF4A stimulated the unwinding activity of Ded1p (*Figure 1E*). Since the weak activity of eIF4A rules out additive effects of the two helicases, the data reveal an impact of eIF4A on the helicase activity of Ded1p, indicating an unwinding-competent eIF4A-Ded1p complex.

The addition of eIF4G to Ded1p and eIF4A further increased the unwinding activity seen with only Ded1p and eIF4A (*Figure 1E*). In contrast, the addition of eIF4G to Ded1p alone decreased the rate constant for Ded1p, as previously reported (*Putnam et al., 2015*). The weak activity of eIF4A with eIF4G precludes additive effects of Ded1p and the eIF4A-eIF4G complex (*Rajagopal et al., 2012*). Our data therefore suggest that Ded1p, eIF4A and eIF4G can function together to unwind RNA duplexes.

Preparations of eIF4G contained the cap binding protein eIF4E (*Hilliker et al., 2011*), which is required for stability of eIF4G in vitro (*Dominguez et al., 2001*). EIF4E had been previously shown to interact with Ded1p, but it did not affect the RNA-stimulated ATPase activity of Ded1p (*Senissar et al., 2014*). Consistent with these results, we did not detect a significant impact of eIF4E on the unwinding activity of Ded1p (not shown). We therefore did not further investigate the impact of eIF4E alone.

## Ded1p is the unwinding module in complexes with eIF4A and eIF4G

We next examined the respective contributions of eIF4A and Ded1p to the unwinding activity when both enzymes were present. We first included hippuristanol in the unwinding reactions (*Figure 2A*,

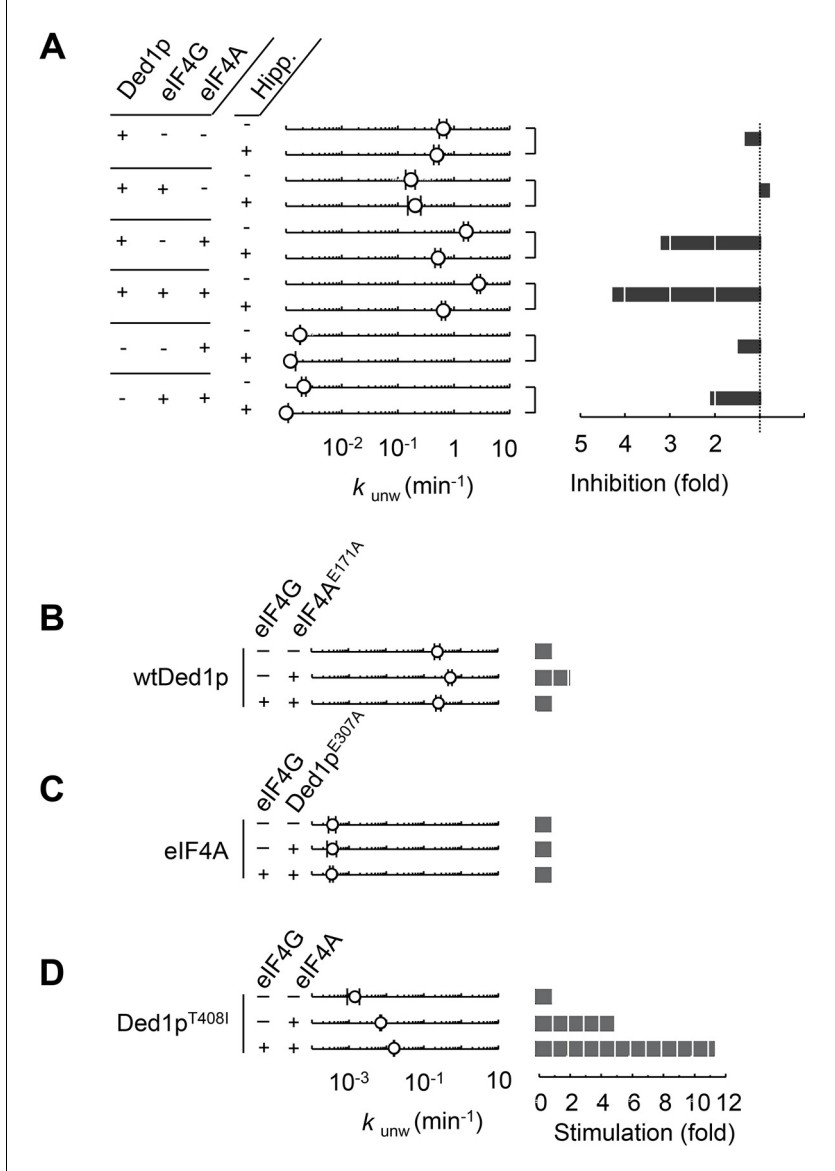

**Figure 2.** Ded1p is the unwinding module in complexes with eIF4A and eIF4G. (**A**) Impact of hippuristanol on of Ded1p, eIF4A, with and without eIF4G, and in combination. Reactions were performed as in **Figure 1E** with 0.2% (v/v) DMSO and 20 μM Hippuristanol, as indicated. *Left panel*: Observed unwinding rate constants ($k_{unw}$). For representative unwinding reactions see **Figure 2—figure supplement 1**. Error bars represent one standard deviation of at least three independent measurements. *Right panel*: decrease or increase of $k_{unw}$ by hippuristanol. (**B**) Impact of eIF4A$^{E171A}$(100 nM) on the unwinding activity of Ded1p (100 nM) with or without eIF4G (100 nM), measured as in **Figure 1E**, but without DMSO. *Left panel*: Observed unwinding rate constants ($k_{unw}$). Error bars represent one standard deviation of at least three independent measurements. *Right panel*: increase of $k_{unw}$, compared to reaction without eIF4A$^{E171A}$. (**C**) Impact of Ded1p$^{E307A}$ on the unwinding activity of eIF4A with or without eIF4G, measured as in **Figure 2B**. *Left panel*: Observed unwinding rate constants ($k_{unw}$). Error bars represent one standard deviation of at least three independent measurements. *Right panel*: increase of $k_{unw}$, compared to reaction without Ded1p$^{E307A}$. (**D**) Impact of eIF4A on of Ded1p$^{T408I}$ with or without eIF4G, measured as in **Figure 2B**. *Left panel*: Observed unwinding rate constants ($k_{unw}$). Error bars represent one standard deviation of at least three independent measurements. *Right panel*: increase of $k_{unw}$, compared to reaction without eIF4A. See also **Figure 2—figure supplement 1**.

The following figure supplement is available for figure 2:

**Figure supplement 1.** Reaction progress curves for **Figure 2A**.

*Figure 2—figure supplement 1*). As expected, hippuristanol decreased unwinding activity for eIF4A alone and in combination with eIF4G, but did not significantly impact unwinding of Ded1p alone or in combination with eIF4G (*Figure 2A*). However, hippuristanol decreased unwinding for Ded1p in the presence of eIF4A, with and without eIF4G, essentially to the level seen with Ded1p alone (*Figure 2A*). The data suggest that Ded1p is the unwinding module in the complexes with eIF4A, but that functional integrity of eIF4A is important for its impact on Ded1p. This observation provides a rationale for the exacerbating effect of hippuristanol on the growth defect of the *ded1-95* mutant strain (*Figure 1B*). To further probe the role of eIF4A in complexes with Ded1p, we tested the impact of a mutation of the ATP binding site of eIF4A (eIF4A$^{E171A}$), which abolishes RNA unwinding, as well as ATP binding and hydrolysis (*Iost et al., 1999*). EIF4A$^{E171A}$ stimulated unwinding by wtDed1p (*Figure 2B*). However, the addition of eIF4G reduced the stimulation to the level roughly seen with Ded1p alone (*Figure 2B*). These observations provide additional evidence that Ded1p is the unwinding module in complexes with eIF4A.

A mutation in the ATP binding site of Ded1p (Ded1p$^{E307A}$), which abolishes RNA unwinding, as well as ATP binding and hydrolysis (*Hilliker et al., 2011*; *Iost et al., 1999*), did not notably alter the basal unwinding activity of eIF4A alone or with eIF4G (*Figure 2C*). This observation indicates that Ded1p dictates the level of activity seen in complexes with eIF4A and eIF4G. Finally, we tested the Ded1p$^{T408I}$ mutation (*Figure 2D*), which showed lower activity than wtDed1p, but higher unwinding activity than eIF4A alone or in complex with eIF4G (*Figure 2B–D*, compare first points). EIF4A stimulated Ded1p$^{T408I}$ and addition of both, eIF4A and eIF4G, stimulated even more (*Figure 2D*). Collectively, these data show that Ded1p determines the level of unwinding activity in complexes with eIF4A, further supporting the notion that Ded1p is the unwinding module in the complexes with eIF4A.

## The N-terminus of Ded1p is critical for interaction with eIF4A

We next asked which region of Ded1p was critical for the interaction with eIF4A. We removed the C-terminus of Ded1p (Ded1p$^{1-535}$), which interacts with eIF4G (*Hilliker et al., 2011*), and measured unwinding activity of Ded1p$^{1-535}$ with eIF4A, eIF4G, or both (*Figure 3A*). As expected, eIF4G alone did not notably impact Ded1p$^{1-535}$, which showed lower unwinding activity than wtDed1p. EIF4A stimulated Ded1p$^{1-535}$, and this stimulation was increased by eIF4G (*Figure 3A*). These observations imply that the C-terminus of Ded1p is not critical for the interaction with eIF4A, even in the presence of eIF4G.

We next measured the impact of eIF4A, eIF4G, or both on Ded1p$^{117-604}$, which lacked the N-terminus. Compared to wtDed1p, Ded1p$^{117-604}$ showed reduced unwinding activity (*Figure 3A*). EIF4G alone slightly stimulated Ded1p$^{117-604}$, consistent with an interaction between eIF4G and Ded1p$^{117-604}$. The slight stimulation of Ded1p$^{117-604}$ by eIF4G, as opposed to the inhibition seen with wtDed1p, likely reflects an increase in RNA affinity of Ded1p$^{117-604}$ by eIF4G. This increase in affinity is also seen with wtDed1p, but does not result in a stimulation of the wild type protein under identical reaction conditions (*Putnam et al., 2015*). Most notably, eIF4A no longer stimulated Ded1p$^{117-604}$, and addition of eIF4G did not increase the activity further (*Figure 3A*). No direct interaction between Ded1p$^{117-604}$ and eIF4A was detected in pull down experiments (*Figure 3—figure supplement 1A*). Together, the data show that the N-terminus of Ded1p is important for the interaction between eIF4A and Ded1p, with and without eIF4G, and further support the notion that eIF4A and Ded1p do not simply act additively.

In the cell, deletion of the N-terminus conferred a severe cold-sensitive growth phenotype (*Figure 3B*), consistent with a prior report (*Banroques et al., 2011*). Notably, overexpression of eIF4A did not alleviate the cold-sensitive growth defect of Ded1p$^{117-604}$ (*Figure 3B*), as it had for another Ded1p mutation (*Figure 1A*). This observation emphasizes the biological significance of the interaction between eIF4A and the N-terminus of Ded1p.

In contrast to the N-terminus, the C-terminus of Ded1p was not critical for interaction with eIF4A, even in the presence of eIF4G. Given that the C-terminus is important for trimerization of Ded1p (*Putnam et al., 2015*), our findings raised the question whether Ded1p in complexes with eIF4A was still able to form trimers. We therefore examined the impact of eIF4A on oligomerization of Ded1p using chemical protein-protein crosslinking (*Figure 3C*). Without eIF4A, trimer formation of Ded1p was seen (*Putnam et al., 2015*). With eIF4A, no Ded1p trimer was observed, and a prominent new species emerged with a molecular weight consistent with a predominant one to one Ded1p-eIF4A

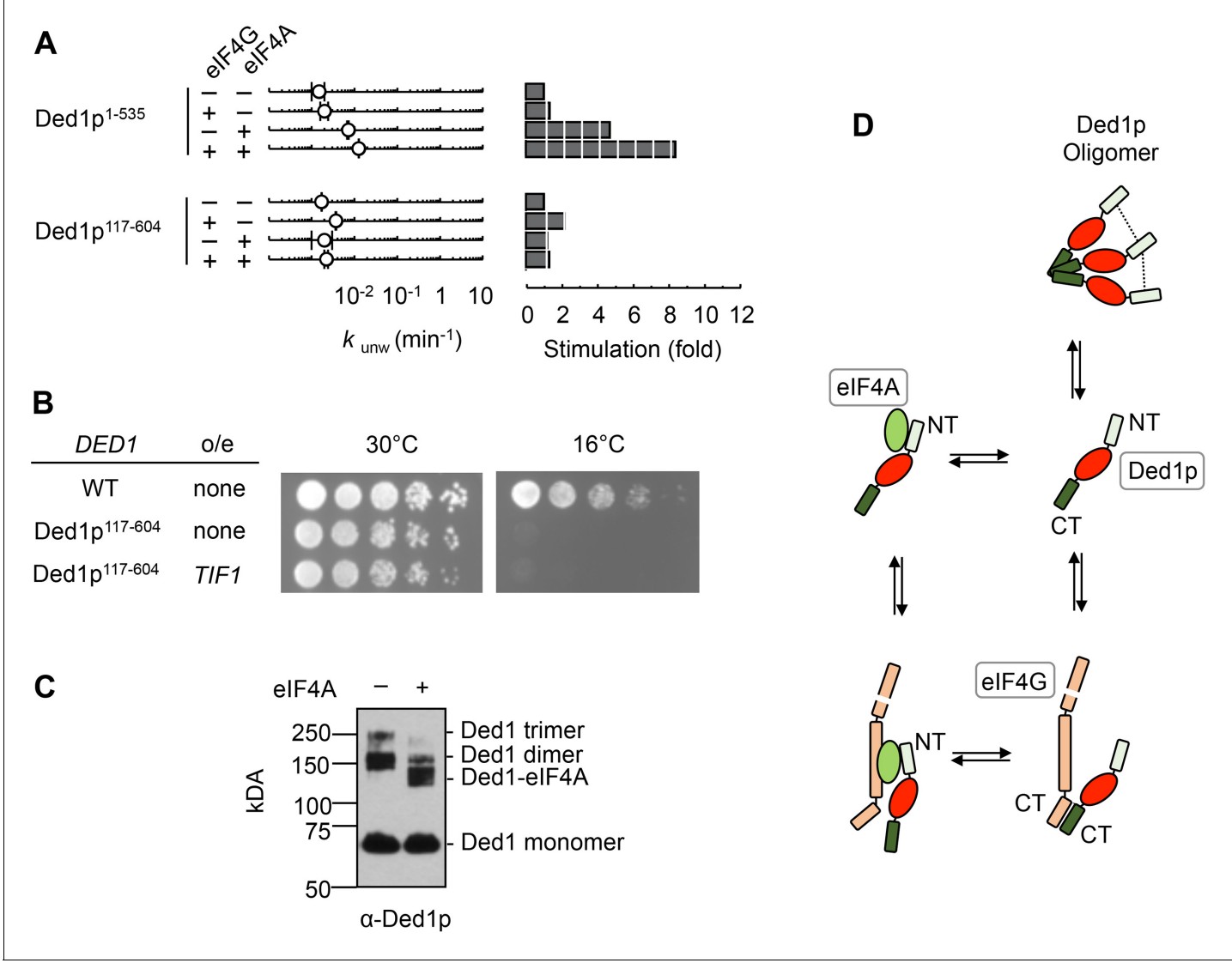

**Figure 3.** The N-terminus of Ded1p is critical for the interaction with eIF4A. (**A**) Impact of eIF4A on of Ded1p[1-535] (C-terminus deleted, upper panels) and Ded1p[117-604] (N-terminus deleted, lower panels) with or without eIF4G, measured as in *Figure 2B*. *Left panel*: Observed unwinding rate constants ($k_{unw}$). Error bars represent one standard deviation of at least three independent measurements. *Right panel*: increase of $k_{unw}$, compared to reactions with Ded1p alone. (**B**) Effect of the deletion of the N-terminus of Ded1p (Ded1p[117-604]), without or with EIF4A (*TIF1*) overexpression (o/e), on growth in yeast at 30°C and 16°C. (**C**) Formaldehyde crosslinking of Ded1p without and with eIF4A in the presence of RNA (16 bp, 25 nt 3'- overhang) and ADPNP. Mobilities of the Ded1p trimer, dimer, monomer, and the Ded1p-eIF4A complex are indicated. Proteins were visualized by western blot (α-Ded1p). (**D**) Schematic model for the basic functional architecture of Ded1p-eIF4A and Ded1p-eIF4A-eIF4G complexes. Ovals represent the helicase cores of Ded1p (red) and eIF4A (green). C- and N-termini of Ded1p and eIF4G are marked. The dotted lines between the N-termini in the Ded1p trimer indicate interactions between the N-termini that contribute to oligomrization. See also *Figure 3—figure supplement 1* and *Supplementary file 1A*.

The following figure supplement is available for figure 3:

**Figure supplement 1.** Direct interaction between eIF4A and the amino terminus of Ded1p.

complex (*Figure 3C*, *Figure 3—figure supplement 1B,C*). The Ded1p dimer species seen in the sample with eIF4A represents remaining Ded1p. The data thus indicate that eIF4A interferes with Ded1p oligomerization, even though the C-terminus of Ded1p is not critical for the complex formation with eIF4A.

Collectively, results shown above and previous data on the Ded1p-eIF4G interaction and Ded1p oligomerization (*Putnam et al., 2015*) suggest a basic model for the functional architecture of the complexes formed between Ded1p, eIF4A, and eIF4G (*Figure 3D*). Without eIF4A, Ded1p interacts via its C-terminus with eIF4G. This interaction precludes formation of a Ded1p trimer (*Putnam et al., 2015*). The main contact between Ded1p and eIF4A involves the N-terminus of Ded1p, even in the presence of eIF4G. EIF4A also interferes with oligomerization of Ded1p.

## A quantitative framework for the interactions between Ded1p, eIF4A, eIF4G, RNA, and ATP

The data above indicated the existence of multiple complexes formed between Ded1p, eIF4A, and eIF4G. To understand biochemical characteristics of these complexes, their interdependencies, and the roles of ATP and RNA, we set out to establish a mechanistic framework describing the thermodynamic stability of these complexes, their RNA and ATP affinities, and their unwinding activities. To our knowledge, no comparable framework exists for any dynamic complex with multiple RNA binding proteins. To devise an instructive quantitative model, it was important to measure the interactions under conditions where ATP binding and turnover as well as RNA binding and remodeling occurred simultaneously. These conditions are best met in unwinding reactions. We therefore performed a large set of unwinding reactions where we systematically varied concentrations of Ded1p, eIF4A, eIF4G, and ATP. Representative data show, among other trends, scaling of the Ded1p stimulation with ATP and Ded1p concentrations, and biphasic response curves with increasing concentrations of eIF4G (*Figure 4—figure supplements 1–4*). The data indicate intricate interdependencies between the three proteins, ATP and RNA.

Using the measured unwinding data, we assembled a quantitative model to describe the thermodynamic stability of all complexes formed between Ded1p, eIF4A, and eIF4G, their RNA and ATP affinities, and their unwinding rate constants (*Figure 4A*). Model building, data fitting strategy and assessment of model quality are described the Materials and methods section, in *Figure 4—figure supplements 5,6* and *Supplementary files 2,3*. The model considers all 28 distinct complexes formed between Ded1p, eIF4A, eIF4G, RNA and ATP (*Figure 4A*), plus complexes formed by Ded1p during oligomerization (*Putnam et al., 2015*). 108 individual interactions were modeled and more than 3000 individual data points were used in the global datafit. Interactions between Ded1p, eIF4A and eIF4G were modeled as one to one interactions, based on the data above (*Figure 3C*), and on previous data characterizing the Ded1p-eIF4G interaction (*Putnam et al., 2015*). The model faithfully recapitulates measured unwinding rate constants (*Figure 4B*, *Figure 4—figure supplement 6*) as well as previously reported ATP affinities for eIF4A and the eIF4A-eIF4G complex (*Rajagopal et al., 2012*; *Mitchell et al., 2010*) (*Supplementary file 2*).

The model reveals the formation of a thermodynamically stable complex between Ded1p, eIF4A, and eIF4G, whose stability further increases with ATP, RNA, or both (*Figure 5A*, *Supplementary file 2F*). The stability of the Ded1p-eIF4A complex, although lower than for the Ded1p-eIF4A-eIF4G complex, also increases with ATP, RNA, or both (*Figure 5A*, *Supplementary file 2D*). In contrast, the Ded1p-eIF4G, and eIF4A-eIF4G complexes are more stabilized by RNA than by ATP (*Figure 5A*, *Supplementary file 2C–E*). To independently verify the calculated high stability of the Ded1p-eIF4G-eIF4A complex, we used microscale thermophoresis (MST) (*Seidel et al., 2013*) to directly measure binding of Ded1p to a pre-formed eIF4G-eIF4A complex in the absence of RNA and ATP. We measured an apparent dissociation constant of $K_{1/2} < 7$ nM (*Figure 5B*), the experimentally accessible lower limit of affinity with the available MST setup. This value is consistent with the corresponding affinity calculated with our model (*Supplementary file 2F*). Given the high thermodynamic stability of the Ded1p-eIF4G-eIF4A complex, we suggest to consider Ded1p, like eIF4A, as integral functional part of eIF4F. We note that high thermodynamic stability does not equal persistence of the complex. In fact, the complex between the three proteins is unlikely to be long-lived, given that both eIF4A and Ded1p constantly turn over ATP.

Our model further revealed that unwinding rate constants at ATP and protein saturation and RNA affinities at ATP saturation are similar for the Ded1p trimer and for the complexes containing Ded1p and eIF4A (*Figure 5C*, *Supplementary file 2A*). Ded1p is stimulated by eIF4A at lower Ded1p concentrations because the complexes with eIF4A bind as single units with hyperbolic binding isotherms, whereas Ded1p oligomerizes cooperatively, which results in a sigmoidal isotherm (*Figure 5D*, *Figure 4—figure supplement 1*). Accordingly, stimulation of Ded1p by eIF4A, with or

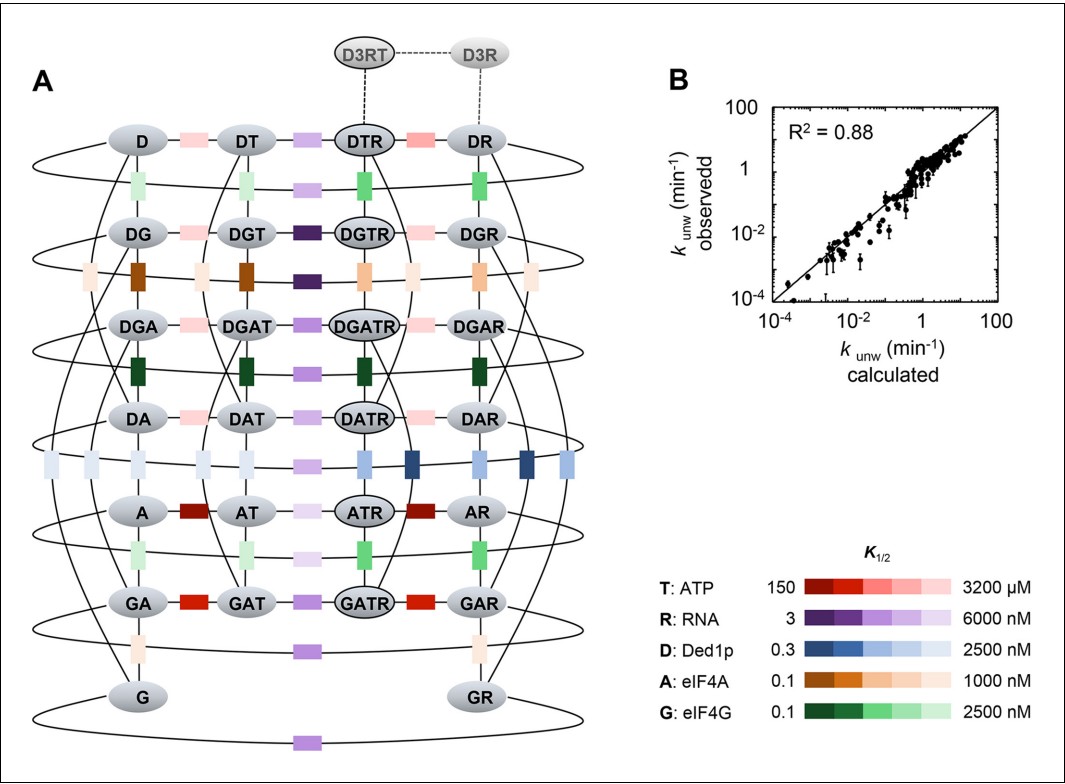

**Figure 4.** Thermodynamic framework for the interactions between Ded1p, eIF4A and eIF4G. (**A**) Graphic representation of the framework. Ovals represent the complexes containing Ded1p (D), eIF4A (A), eIF4G (G), RNA (R), ATP (T). D3 indicates the Ded1p timer (**Putnam et al., 2015**). Black lines mark a transition from one complex to another through binding of one of the components. Colored squares indicate the calculated equilibrium dissociation constant ($K_{1/2}$) for the transition, as shown in the legend at the lower right. For visual reasons, the framework does not depict specific states for ATP-bound eIF4A. Equilibrium constants for all individual transitions are listed in **Supplementary file 2**. (**B**) Observed apparent unwinding rate constants versus those calculated with the model in panel (**A**). Measured and calculated rate constants from different reaction conditions are listed without reference to the specific reaction conditions, solely to visualize the overall agreement between observed and calculated values. For fitting parameters see **Figure 4—figure supplement 5** and **Supplementary file 2**. The black line represents a diagonal, $R^2$ is the correlation coefficient for a linear fit of the shown data to this line. Error bars mark the standard deviation for the observed values. See also **Figure 4—figure supplements 1–6** and **Supplementary files 2,3**.

The following figure supplements are available for figure 4:

**Figure supplement 1.** Impact of the Ded1p concentration on the modulation of its unwinding activity by eIF4A and eIF4G.

**Figure supplement 2.** Impact of eIF4A concentration on the modulation of Ded1p unwinding activity by eIF4A and eIF4G.

**Figure supplement 3.** Impact of eIF4G concentration on the modulation of Ded1p unwinding activity by eIF4A and eIF4G.

**Figure supplement 4.** Impact of ATP concentration on the modulation of Ded1p unwinding activity by eIF4A and eIF4G.

**Figure supplement 5.** Relative Chi square values ($X^2$) for variations of two representative parameters.

**Figure supplement 6.** The thermodynamic model adequately describes observed data for each subset of complexes.

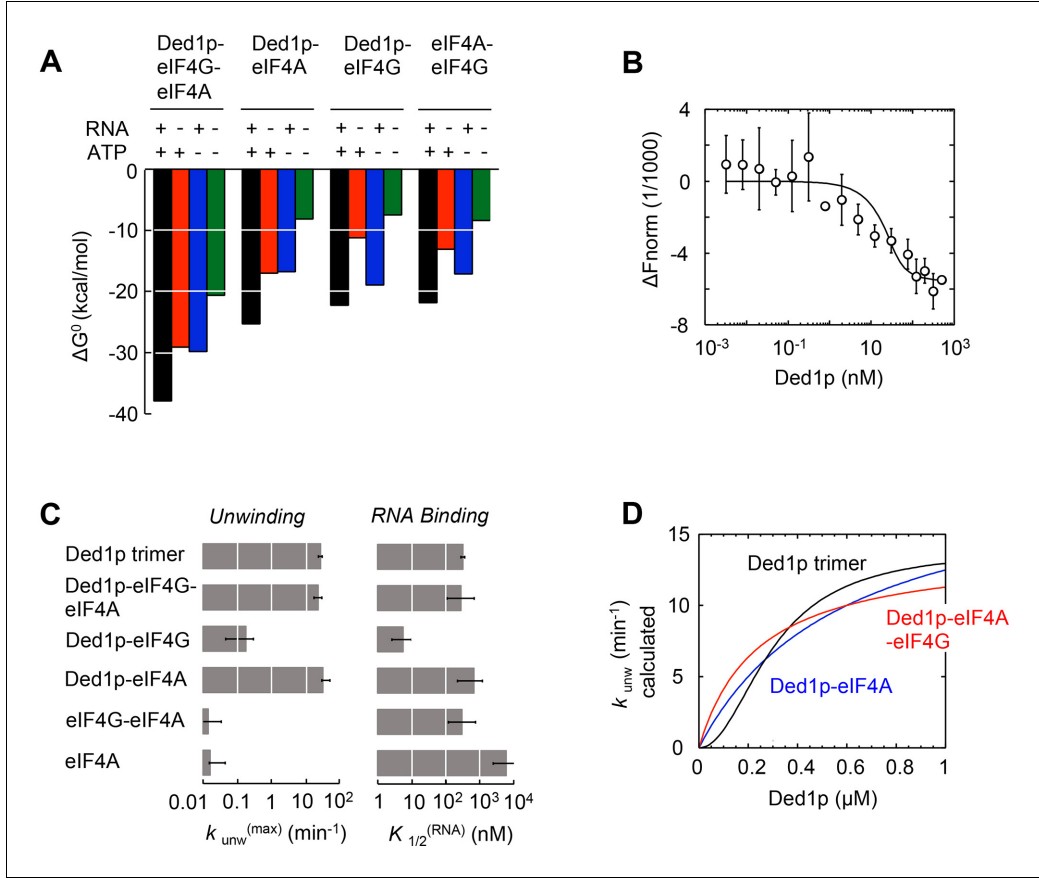

**Figure 5.** Functional parameters of complexes with Ded1p, eIF4A, or both. (A) Free energies ΔG° for complexes formed between Ded1p, eIF4A, eIF4G, RNA (25 nt 3'-overhang, 16 bp duplex) and ATP in the combinations shown, calculated with the model in *Figure 4A*. (B) MST measurement of Ded1p binding to eIF4A-eIF4G (30 nM eIF4G, 36 nM Cy5-labeled eIF4A, increasing concentrations of Ded1p as indicated, 25 µg/mL RNase A, 23°C). Datapoints show the change in fluorescence signal at the indicated Ded1p concentrations, normalized to the signal without Ded1p. Error bars indicate one standard deviation of multiple independent measurements. Datapoints were fitted with the quadratic form of the binding isotherm ($K'_{1/2} < 6.6$ nM). (C) Unwinding rate constants ($k_{unw}^{(max)}$, ATP and protein saturation, RNA: 25 nt 3' overhang, 16 bp duplex) and RNA affinity ($K_{1/2}^{(RNA)}$, ATP saturation) for complexes formed between Ded1p, eIF4A, eIF4G, and, for comparison, for Ded1p and eIF4A, calculated with the model in *Figure 4A*. (D) Unwinding rate constants (RNA: 25 nt 3' overhang, 16 bp duplex) for the Ded1p timer, the Ded1p-eIF4A, and the Ded1p-eIF4A-eIF4G complex as function of the Ded1p concentration (ATP saturation, eIF4 A = 2000 nM, eIF4G = 100 nM), calculated with the model in *Figure 4A*. See also *Figure 5—figure supplement 1*.

The following figure supplement is available for figure 5:

**Figure supplement 1.** ATPase activity of complexes formed between Ded1p, eIF4A, and eIF4G at RNA and ATP saturation.

without eIF4G, decreases at higher Ded1p concentration (*Figure 4—figure supplement 1*). The similar unwinding rate constants at ATP and protein saturation for the Ded1p trimer, the Ded1p-eIF4A and the Ded1p-eIF4A-eIF4G complexes are consistent with the notion that a Ded1p protomer acts as the unwinding unit in the complexes with eIF4A. ATP turnover by Ded1p was not significantly affected by eIF4A and eIF4G (*Figure 5—figure supplement 1*). The observations suggest that eIF4A and eIF4G function to recruit Ded1p as RNA remodeling unit.

## Modulation of Ded1p activity by eIF4A and eIF4G is impacted by substrate architecture

Having dissected biochemical interdependencies between Ded1p, eIF4A and eIF4G, we next tested whether and how the substrate architecture (lengths of duplexes, lengths and orientation of unpaired tails) impacts the degree by which eIF4A, eIF4G, or both, modulate the unwinding activity of Ded1p. We first performed unwinding reactions with substrates containing different overhang orientations (*Figure 6A*). We found a modestly enhanced preference of 3' vs. 5' tailed substrates for Ded1p with eIF4A and eIF4G (*Figure 6A*). This observation is notable, because eIF4G imparts on eIF4A alone a preference for 5' tailed substrates (*Rajagopal et al., 2012*) (*Figure 6—figure supplement 1*).

We next examined the impact of the length of the unpaired 3' tail (*Figure 6B,C*; *Figure 6—figure supplement 2*). To this end, we obtained functional affinities for the various protein complexes for a substrate with 10 nt 3' overhang (*Figure 6B*). These parameters were calculated from sets of unwinding reactions with the 10 nt tailed substrate with varying concentrations of Ded1p, eIF4A, and eIF4G, analogous to the process described for the substrate with the 25 nt overhang (*Figure 6—figure supplement 3*). We then compared the functional affinities for the 10 and the 25 nt tailed substrates for each protein complex (*Figure 6B*). For Ded1p alone, affinity increased for a 25 nt vs. a 10 nt tail (*Figure 6B*, lower panel), consistent with previous observations (*Yang et al., 2007a*). Addition of eIF4A, eIF4G, or both, increased the relative affinity of the complexes for 25 nt vs. 10 nt tails (*Figure 6B*). Yet, at low Ded1p concentrations, eIF4A and eIF4G stimulated Ded1p unwinding stronger for a substrate with a 10 nt tail (*Figure 6—figure supplement 2*), compared to a substrate with a 25 nt tail (*Figure 6A*). The stronger stimulation for substrates with shorter tails arises from the hyperbolic shape of the functional binding isotherm of the Ded1p-eIF4G-eIF4A complex, compared to the sigmoidal isotherm for the Ded1p trimer (*Figure 6C*). These data show that activity stimulation can scale with the concentration of components, without a marked change in affinity.

Finally, we examined the impact of duplex length on the degree by which eIF4A, eIF4G, or both, modulated the unwinding activity of Ded1p (*Figure 6D*; *Figure 6—figure supplement 2*). Since duplex length impacts only the unwinding rate constant, but not the substrate affinity (*Putnam et al., 2015*), we calculated unwinding rate constants at ATP and enzyme saturation for duplexes of different length using our thermodynamic model (*Figure 4*), and data collected from substrates with 13, 16, and 19 basepairs and identical 3', 25 nt overhangs (*Figure 6D*). The shortest duplex was unwound fastest by the eIF4A-Ded1p complex, and slowest by the eIF4A-eIF4G-Ded1p complex. For the longest duplex, this order was reversed (*Figure 6D*). The scaling of the unwinding rate constants with duplex length correlates with the process of strand separation (*Putnam et al., 2015*). The data thus indicate that both eIF4A and eIF4G directly affect the strand separation step by the Ded1p protomer. EIF4A most likely increases the time by which partially opened duplex stands are kept apart, while decreasing the number of basepairs opened per strand separation event. In contrast, eIF4G and eIF4A together appear to decrease the time by which partially opened duplex stands are kept apart, while increasing the number of basepairs opened per strand separation event. For substrates containing a 5' tail, no comparable scaling of the stimulation with the duplex length was seen, although the stimulation was somewhat higher for the longer duplex (*Figure 6A*, *Figure 6—figure supplement 2*). Collectively, our data show that substrate architecture influences the degree by which unwinding activity of Ded1p is impacted by eIF4A and eIF4G, alone and in combination. Our observations further reveal that eIF4A and eIF4G directly influence the strand separation step of Ded1p (*Figure 6D*), and that substrate affinity of each protein complex scales with the length of the unpaired tail (*Figure 6B,C*).

## Discussion

In this study, we demonstrate direct and functional interactions between the DEAD-box helicases eIF4A and Ded1p. Although both helicases have been firmly implicated in translation initiation, they had not been viewed as physically connected. Our data further show that eIF4A and Ded1p simultaneously interact with eIF4G to form a thermodynamically stable complex. The findings reveal a physical and functional connection between Ded1p, eIF4A and eIF4G, and suggest that Ded1p is an integral functional part of the eIF4F complex.

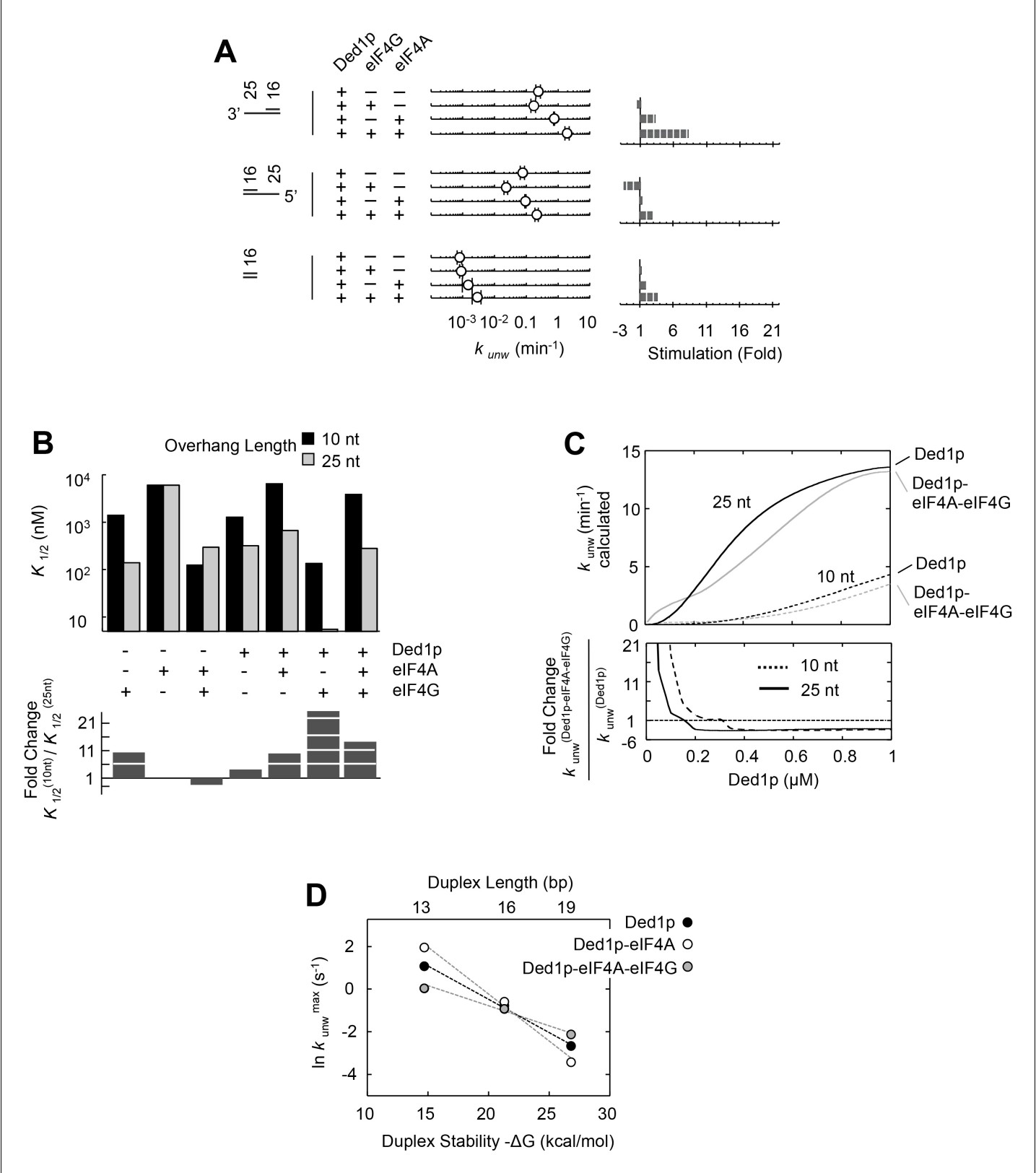

**Figure 6.** Impact of the substrate architecture on the functional parameters of complexes with Ded1p, eIF4A, and eIF4G. (**A**) Impact of eIF4A, eIF4G, or both, on the unwinding activity of Ded1p with 5' and 3' tailed substrates (16 bp, 25 nt 3'- or 5'-tail), and blunt-end, shown schematically at the left (numbers indicate nucleotides in unpaired overhang and basepairs in duplex region). Reactions were performed as in *Figure 2B*. Sliding point graphs
*Figure 6 continued on next page*

*Figure 6 continued*

show observed unwinding rate constants ($k_{unw}$, average of at least three independent measurements), error bars mark one standard deviation. Bar graphs mark increase or decrease of $k_{unw}$, compared to the reaction with only Ded1p for each substrate. (B) *Upper panel*: Affinities ($K_{1/2}$) for RNA substrates with 10 nt and 25 nt unpaired regions 3' to the duplex (16 bp) for Ded1p, eIF4A, eIF4G alone and in combination as indicated, calculated with the model shown in *Figure 4A*, and with data obtained with two substrates containing either 10 or 25 nt overhangs. For modeling parameters see *Figure 6—figure supplements 2,3*, *Supplementary file 2* and Materials and methods. *Lower panel*: Fold change in $K_{1/2}$ between the substrates with 10 and 25 nt unpaired regions. (C) *Upper panel*: Apparent unwinding rate constants ($k_{unw}$, ATP = 4 mM, eIF4A = 100 nM, eIF4G = 100 nM) for Ded1p alone and for the Ded1p-eIF4A-eIF4G complex on RNA substrates with 10 or 25 nt overhangs, 3' to a 16 bp duplex, as function of increasing Ded1p concentration, calculated with the model shown in *Figure 4A*, and with data obtained with two substrates containing either 10 or 25 nt overhangs. *Lower panel*: Fold change in $k_{unw}$ for reactions with Ded1p-eIF4A-eIF4G, compared to reactions with Ded1p alone for both substrates. (D) Unwinding rate constants at ATP and protein saturation for RNA substrates with 25 nt 3' overhangs and 13, 16, and 19 bp duplexes, for Ded1p, the Ded1p-eIF4A complex and the Ded1p-eIF4A-eIF4G complex, calculated with the model shown in *Figure 4*. For modeling parameters see *Figure 4—figure supplements 5,6*; *Figure 6—figure supplement 2*, *Supplementary files 2,3* and Materials and methods. See also *Figure 6—figure supplements 1,3*.

The following figure supplements are available for figure 6:

**Figure supplement 1.** Stimulation of unwinding activity of eIF4A by eIF4G with an RNA with a 5' tail.

**Figure supplement 2.** Substrate architecture affects the modulation of Ded1p activity by eIF4A and eIF4G.

**Figure supplement 3.** Correlation between observed and calculated unwinding rate constants ($k_{unw}$) for the 16 bp substrate with 10 nt 3' tail.

Our data show that Ded1p, eIF4A and eIF4G, all of which bind RNA, can form four discrete complexes (Ded1p-eIF4A, Ded1p-eIF4G, Ded1p-eIF4A-eIF4G, eIF4A-eIF4G) with defined biochemical properties. Most importantly, Ded1p is the primary RNA remodeling unit in all complexes where it is present. Compared to previously described complexes between eIF4A, eIF4G, and eIF4B, the complexes with Ded1p display orders of magnitude more potent RNA remodeling activity than complexes that only contain eIF4A. This observation raises the possibility that remodeling of stable RNA structures in cellular 5'UTRs, which is often attributed to eIF4A, might in fact be carried out by complexes that contain Ded1p.

Our data suggest that the various complexes readily form at physiological concentrations of protein, RNA and ATP. Affinities determined with our model and reported cellular concentrations of the proteins, RNA and ATP (*Firczuk et al., 2013*; *Koç et al., 2004*; *Zenklusen et al., 2008*), allowed us to estimate the prevalence of all complexes at concentrations around the physiological concentrations of Ded1p and eIF4G (*Figure 7*). These calculations suggest that all complexes exist under physiological conditions, thus rationalizing reported co-immunoprecipitation data (*Senissar et al., 2014*). The calculations further show that at the reported cellular concentrations of Ded1p and eIF4G (~1 μM), the majority of the two proteins is bound in the Ded1p-eIF4A-eIF4G complex (*Figure 7*). Almost all of the remaining Ded1p and remaining eIF4G is bound to eIF4A, and only little Ded1p oligomer is present.

Variations of both, Ded1p and eIF4G concentrations, even by a factor of only two, significantly alter the prevalence of the respective complexes, with strikingly distinct functional impact for Ded1p and eIF4G. For Ded1p, the changes in the complex ratio have only little impact on helicase activity, since complexes with Ded1p have similar unwinding activities and RNA affinities, although this depends somewhat on the duplex length (*Figure 6B,D*). Increasing eIF4G concentrations dramatically decreases the fraction of eIF4G that is associated with Ded1p and thus with potent remodeling activity. These observations suggest that Ded1p, in addition to eIF4A, might be important for the function of eIF4G. The data provide a possible rationale for the roughly equal cellular concentration of eIF4G and Ded1p (*Firczuk et al., 2013*).

The Ded1p-eIF4A-eIF4G complex is thermodynamically most stable, suggesting an evolutionary adaptation to favor formation of this complex. For this reason, we suggest to consider Ded1p, like eIF4A, as integral part of eIF4F. Our data highlight a previously unknown role of eIF4A as a co-factor for Ded1p. This function is consistent with recently discovered overlapping roles of Ded1p and eIF4A in translation initiation (*Sen et al., 2015*), and with the partial complementation of *ded1* mutations by overexpression of eIF4A (*Figure 1A*). Since the cellular eIF4A concentration exceeds the

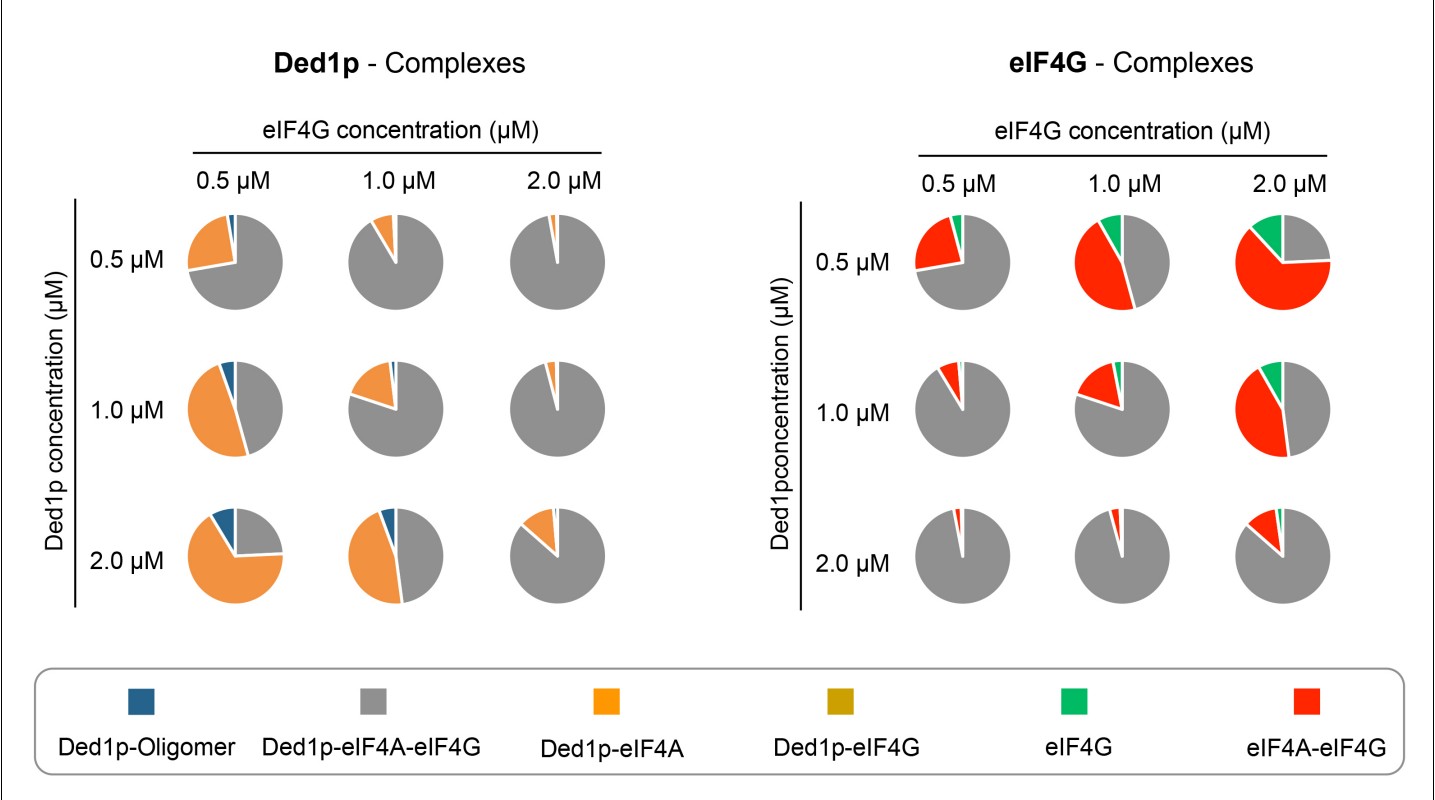

**Figure 7.** Calculated prevalence of complexes formed between Ded1p, eIF4A and eIF4G at physiological protein concentrations. Fraction of Ded1p (left) and eIF4G (right) present in the various complexes with eIF4A, eIF4G, or both, at three different Ded1p and eIF4G concentrations around their reported physiological concentrations of ~1 µM (RNA = 6 µM, ATP = 2.5 mM). Relative abundance of the complexes were calculated with the binding parameters obtained from the thermodynamic framework (*Figure 4A*, *Supplementary file 2*). See also *Figure 4* and *Supplementary file 2*.

Ded1p concentration by a factor of approximately five, our findings do not exclude roles of eIF4A in translation initiation that are independent of Ded1p.

Our results define a basic functional architecture of the complexes between Ded1p eIF4A and eIF4G (*Figure 3D*). Most notably, Ded1p utilizes distinct regions to bind to eIF4A and eIF4G. If eIF4A is present, the Ded1p N-terminus is most critical for binding, whereas the Ded1p C-terminus is critical for binding to eIF4G alone. eIF4G and eIF4A allosterically affect the unwinding activity of the Ded1p protomer. In energetic terms, these effects are smaller than the allosteric impact of co-factors on RNA helicases seen in other multi-component complexes, such as the TRAMP complex (*Jia et al., 2011*). However, Ded1p allosterically impacts eIF4A, eIF4G, or both, as reflected in the marked increase in RNA affinity for the two proteins. This increase in the RNA affinity most likely indicates a function of the proteins as RNA binding adaptors for the Ded1p protomer that separates the strands. Unwinding activity and RNA affinity in the Ded1p-eIF4A, and Ded1p-eIF4A-eIF4G complexes depends on the substrate architecture to a moderate degree, but substrate preferences of Ded1p are not drastically altered by eIF4A or by eIF4A-eIF4G.

The intricate connection between Ded1p, eIF4G and eIF4A implies that removal of Ded1p or disruption of the interaction between Ded1p and eIF4A profoundly affects RNA remodeling by complexes that contain eIF4A. In addition, complexes containing Ded1p are more sensitive to ATP depletion and inhibition by AMP than complexes containing only eIF4A, because Ded1p has a weaker ATP affinity than eIF4A (*Putnam et al., 2015*; *Putnam and Jankowsky, 2013*; *Rajagopal et al., 2012*). Complexes with Ded1p might therefore be more sensitive to the metabolic state of the cell than complexes with only eIF4A (*Putnam and Jankowsky, 2013*). A broad regulatory potential is thus likely to be associated with the interaction between Ded1p to eIF4A. Consistent with this hypothesis, the Ded1p N-terminus that binds to eIF4A, harbors a multitude of proven

and potential sites for post-translational modifications (*Sharma and Jankowsky, 2014*). Modification of these sites could affect the interaction with eIF4A.

Finally, our quantitative analysis of dynamic complexes formed by multiple RNA binding proteins emphasizes that the effects of these proteins on each other cannot simply be described as 'stimulating' or 'inhibiting'. Because multiple sub-complexes are often simultaneously present, both in vitro and in the cell, observed effects depend on the concentration of individual proteins, relative to each other and in absolute molar terms, and on ATP and RNA concentrations. For these reasons, frameworks such as the one developed in this study are required to adequately describe biochemical features of individual complexes, and to define the interactions between all factors, especially if the complexes are not long lived. The approaches outlined in this work could therefore be useful for the quantitative analysis of other dynamic complexes that contain multiple RNA-binding proteins.

# Materials and methods

## Recombinant proteins

### Ded1p and mutant Ded1p proteins

Wild type Ded1p with an N-terminal His$_6$-tag (plasmid: pET-22b(+)-*DED1*, *Supplementary file 1C*) was expressed in *E. coli* (BL21) and purified and characterized as described (*Yang and Jankowsky, 2005*).

The plasmid for recombinant Ded1p$^{T408I}$ with N-terminal His$_6$-tag (pEJ13, *Supplementary file 1C*) was generated by site-directed mutagenesis of pET-22b(+)-*DED1* with primers X11 and X12 (*Supplementary file 1B*). The presence of the *ded1-95* allele in the transformants was probed for by amplifying plasmid DNA with primers X53 and X54 and subsequent restriction fragment length analysis with DdeI (New England Biolabs, Ipswitch, MA) according to the manufacturer's suggestions. The restriction enzyme cuts only the WT allele but leaves the *ded1-95* allele uncut. WT controls were included in the assay to ensure the validity of the reaction conditions. Ded1p$^{T408I}$ was expressed in *E. coli* (BL21) and purified and characterized as described (*Yang and Jankowsky, 2005*).

The plasmid for Ded1p$^{E307A}$ (N-terminal His$_6$-tag, pET-22b(+)-*DED1*$^{E307A}$, *Supplementary file 1C*) was a gift from Dr. Patrick Linder (Geneva, Switzerland, *Iost et al., 1999*). Ded1p$^{E307A}$ was expressed in *E. coli* (BL21) and purified and characterized as described wt Ded1p described (*Yang and Jankowsky, 2005*).

To produce recombinant HA-tagged Ded1p (Ded1p-HA, N-terminal His$_6$-tag and C-terminal HA$_3$-tag, plasmid:pET-22b(+)-His-Ded1-HA), *DED1* from yeast genomic DNA was amplified by PCR with primers zfDD10 and zfDD11 (*Supplementary file 1B*). The PCR product was gel purified, digested with HindIII and PstI, and ligated into the plasmid pBlueScript II KS(+). Tandem triple HA tags were inserted with PCR amplification of the new plasmid with the primers zfDD12 and zfDD13 (*Supplementary file 1B*). The resulting pBS II KS(+)-DED1-HA plasmid was digested with NheI and HindIII and the ~2.0 kb fragment containing the coding sequence for His$_6$-Ded1p-HA$_3$ was gel purified. The pET-22b(+)-*DED1* plasmid was digested with NheI and HindIII and the ~7.0 kb product containing the backbone of pET-22b(+) was gel purified. The two gel-purified fragments were ligated to form pET-22b(+)-His-Ded1-HA. The coding sequence in the plasmid was verified by sequencing. The plasmid was transfected into *E.coli* (BL21), and recombinant Ded1p-HA was expressed, purified, and characterized as wild type Ded1p.

Ded1p$^{1-535}$ (C-terminus deleted, pET-22b(+)-His- *DED1*$^{1-535}$) was expressed in *E. coli* (BL21) and purified and characterized as described (*Hilliker et al., 2011*).

To produce recombinant Ded1p$^{117-604}$ (N-terminus deleted) the fragment for *DED1*$^{117-604}$ was amplified with the primers zfDD14 and zfDD15 (*Supplementary file 1B*), using the plasmid pET-22b(+)-*DED1* as PCR template. The PCR product was gel purified, digested with XhoI and SacI, and ligated into the pET-22b(+)-His$_6$-thrombin-SUMO plasmid (gift from Dr. Derek Taylor, Case Western Reserve University, Cleveland, OH) at the 3'-end of SUMO sequence. The coding sequence in the resulting plasmid (pET-22b(+)-SUMO- *DED1*$^{117-604}$) was verified by sequencing. Subsequently, the plasmid was introduced into *E.coli* (DE3). Expressed His-SUMO-Ded1p$^{117-604}$ protein was purified through a Nickel column (PrepEaseHis-tagged protein purification resin high specificity, USB), and the eluted protein was incubated with His-tagged SUMO protease at 8°C overnight in order to

remove the His-SUMO-tag. The solution was briefly incubated with Nickel resin to absorb the His-SUMO tag and the His-tagged SUMO protease. Ded1p[117-604] in the supernatant was further purified through a P11 phosphocellulose column (Whatman). Protein was stored in 50 mM Tris-HCl (pH 8.0), 2 mM DTT, 1 mM EDTA, 0.1% (v/v) Triton-X 100, 30% (v/v) glycerol and 300 mM NaCl. Protein homogeneity and concentration were determined by SDS-PAGE and Coomassie Blue staining, with diluted BSA (Roche) and wild-type Ded1p in known concentrations as standards.

### eIF4G protein
Expression and purification of eIF4G [(pET41a(+)-GST-S•Tag-eIF4G1-His) and (pET21b(+)-eIF4E)] were previously described (*Hilliker et al., 2011*).

### eIF4A and mutant eIF4A proteins
pET-28a(+) plasmid encoding an N-terminally His$_6$-tagged wild-type yeast eIF4A (pET-28a(+)-eIF4A) was a gift from Dr. Michael Altmann (Bern, Switzerland) (*Schütz et al., 2008*). To generate plasmid pET-28a(+)-eIF4A$^{E171A}$, two pairs of primers (zfDD16, zfDD17, zfDD18 and zfDD19) were used to introduce the E171A mutation into plasmid pET-28a(+)-eIF4A by the SLIM-PCR procedure according to (*Chiu, et al., 2004*, *2008*). eIF4A and eIF4A$^{E171A}$ were purified as described (*Hilliker et al., 2011*), and stored in 50 mM Tris-HCl (pH 8.0), 2 mM DTT, 1 mM EDTA, 0.1% Triton-X 100, 30% glycerol and 200 mM NaCl.

To create the plasmid for the recombinant GST-eIF4A purification, the eIF4A coding sequence was amplified from pET-28a(+)-eIF4A with the primers zfDD20 and zfDD21, subsequently was digested with SacII and XhoI, and ligated into the pET-41a(+) plasmid to generate the pET-41a(+)-eIF4A plasmid (*Supplementary file 1C*). GST-His$_6$-eIF4A was expressed in *E. coli* (DE3). The protein was purified through a Nickel column (USB), a glutathione sepharose 4B column (GE health care) and a size-exclusion NAP-25 column (BioRad). Purified protein was stored in 50 mM Tris-HCl (pH 8.0), 2 mM DTT, 1 mM EDTA, 0.1% Triton-X 100, 30% glycerol and 200 mM NaCl.

## RNA oligonucleotides
RNA oligonucleotides were purchased from DHARMACON. RNAs were radiolabeled and duplex substrates were prepared and characterized as described (*Yang and Jankowsky, 2005*). For sequences see *Supplementary file 1A*.

## Complementation of *ded1-tam* by eIF4A overexpression
Yeast strains are listed in *Supplementary file 1*. A high copy suppressor screen of cold sensitive *ded1-tam* allele (*Hilliker et al., 2011*) was performed using a yeast genomic tiling library of overexpression vectors (*Jones et al., 2008*; GE Healthcare #YSC5103). The putative suppressors included *DED1* or *TIF2* within the 10-kb genomic inserts. Plasmids containing either wild type *DED1* (pRP1555) or the mutant *ded1-tam* (pRP048) were transformed into a *DED1* yeast shuffle strain containing a chromosomal deletion of *DED1* and wild type copy of *DED1* on a *URA3*-marked plasmid (yRP2799; *Hilliker et al., 2011*). Transformants were grown on selective media before counter-selection against the *URA3*-marked plasmid on 5-fluoroorotic acid (5-FOA; *Boeke et al., 1987*). These strains, now containing the newly transformed copy of *ded1* as an exclusive copy, were transformed with either an empty vector (pGP564; GE Healthcare #YSC5034) or a plasmid overexpressing either *DED1*, *TIF1*, or *TIF2* (pAKH776, 757, 775, or 756) on a *LEU2*-marked vector. Transformants were grown on selective media with 2% dextrose. Cells were grown to mid-log phase in selective media with 2% glucose. The cultures were pin replicated at identical concentrations onto YPDA media and incubated at 30 or 16°C. All of the strains grew like wild type at 30°C.

High copy plasmids containing either *TIF1*, *TIF2*, and *DED1* ORFs and roughly 500 nucleotides up- and downstream of the ORF were made via recombination in yeast. *TIF2*, and *DED1* genes were amplified from the genomic tiling plasmids that suppressed the *ded1* mutant alleles (GE Healthcare #YSC5103). The *TIF1* gene was amplified from genomic DNA of the BY4741 *S. cerevisiae* strain. Primers (*Supplementary file 1B*) contained a 40 nt overhang complementary to the ends of the insertion site in the vector backbone (pGP564). The vector was linearized by digestion with BamHI and XmaI. Gel purified PCR product and linearized vector were transformed into BY4741 and plated on media selective for the circularized vector. Circularized plasmids were isolated from yeast via the

Zymoprep Yeast Plasmid Miniprep II kit (Zymo Research, #D2004) and transformed into DH5α competent *E. coli* (NEB, #C2987) to amplify plasmids. Plasmid inserts were confirmed by sequencing the entire insert.

## Growth measurements with hippuristanol

For growth studies with hippuristanol, a plasmid with the *ded1-95* allele (Ded1p$^{T408I}$, pEJ4, *Supplementary file 1C*) was created. Genomic DNA was amplified with primers X7 and X8 from a yeast strain containing the WT copy of the HIS3 gene (yJC300, gift from Dr. Jeff Coller, Cleveland, OH) and subsequently subcloned (Invitrogen, Carlsbad, CA). The *ded1-95* allele was introduced into pEJ4 by site-directed mutagenesis with primers X11 and X12 generating pEJ6. pEJ4 and pEJ6 were linearized and used to transform BY4741 by standard lithium chloride transformation yielding yeast strains yEJ1 and yEJ3, respectively. The presence of the *ded1-95* allele in the yeast strain was verified by amplifying genomic DNA and subsequent restriction fragment length analysis as described above for pEJ13.

Hippuristanol was a gift from Dr. Jerry Pelletier (McGill University, Canada). WT and *ded1-95* yeast strains were grown in YPD at 30°C, diluted (V = 1.8 mL) to 0.1 $OD_{600}$ and grown for an additional 3 hr to 0.4 $OD_{600}$. Cultures were shifted to 37°C, and grown for 45 min at 37°C. Hippuristanol in DMSO was added to final concentrations of 1 µM, 5 µM, 10 µM and 20 µM. For control reactions, DMSO alone (0.2% v/v) was added. Cells were further grown for 45 min at 37°C. Growth was subsequently followed for 2 hr by measuring $OD_{600}$. The growth rate is reported, normalized to the growth rate of the corresponding strain without Hippuristanol.

## Complementation with *DED1*$^{117-604}$

The plasmid for complementation with *DED1*$^{117-604}$ (N-terminus deleted, plasmid pEJ10) was constructed from pRP1555 (HIS3/CEN) (*Hilliker et al., 2011*), using primers Ded116delFt/Rs and Ded116delRt/Fs (*Supplementary file 1B*). The plasmid insert was confirmed by sequencing the insert. The *ded1*-null strain yRP2799 (*MATa ura3Δ0 leu2Δ0 lys2Δ0 his3Δ1 met15Δ0 ded1::KANMX* pRP1560) was transformed with HIS3-marked wild type *DED1* (pRP1555), or *DED1*$^{117-604}$ (pEJ10). As an additional control we also transformed yRP2799 with an empty HIS3-marked plasmid (pRS413). Transformants were streaked on media lacking histidine and further screened on a 5-FOA containing plate. These strains were transformed with either empty vector (pGP564; GE Healthcare #YSC5034) or a plasmid overexpressing *TIF1* (pAKH775) on a *LEU2*-marked vector. Transformants were grown on selective media with 2% dextrose. Cells were grown to mid-log phase in selective media with 2% glucose. The cultures were pin replicated at identical concentrations onto YPDA media and incubated at 30 or 16°C. At 37°C, no significant growth differences between wt*DED1* and *DED1*$^{117-604}$ were observed.

## Pull-down assays

In vitro pull-down measurements were essentially performed as described (*Putnam et al., 2015*). For the pull-down assays with immobilized eIF4G, 5 µL bed volume of DynaBeads Protein G (Thermo-Fisher Scientific # 10003D) was pre-equilibrated with 200 µL helicase reaction buffer [40 mM Tris–HCl (pH 8.0), 0.5 mM $MgCl_2$, 2 mM DTT, 0.01% (v/v) IGEPAL CA-630] complemented with 100 mM NaCl (HRB-Na100 buffer) at room temperature for 30 min. The pre-equilibrated beads were then incubated with 4 µg anti-S•Tag monoclonal antibody (Novagen, Cat. # 71549) and 420 µg BSA (Roche) in a 25 µL reaction at 8°C overnight. Protein binding reactions (100 µL) included 300 nM S•Tag-eIF4G, 300 nM Ded1p, 1500 nM eIF4A, 20 µg/mL RNases A,B (Roche), as indicated. 3 µL of the protein reaction volume was removed and used as input. The remainder of the reaction volume was rotated at 8°C overnight, and subsequently mixed with the 25 µL equilibrated beads. After further incubation of one hour with rotation, the beads were washed four times with RIPA buffer (50 mM Tris, pH 7.5, 1% IGEPAL CA-630, 0.25% sodium deoxycholate, 0.1% SDS, 200 mM NaCl, 1 mM EDTA, 1 mM PMSF). Proteins were eluted with Laemmli buffer (Biorad), and denatured at 95°C for 3 min. Proteins were resolved on the NuPAGE Novex 4–12% Bis-Tris protein gels (1.0 mm) in MOPS buffer (pH7.7, Life Technology), and visualized by Western blot with primary antibodies against Ded1p (α-Ded1p, polyclonal antibody, raised against full length Ded1p, dilution 1:6000), monoclonal

anti- S•Tag antibody (Novagen, cat. # 71549), and anti-eIF4A polyclonal antibody (Santa Cruz, prod. # sc-27227).

For the pull-down assays with immobilized Ded1p-HA, 1200 nM Ded1p-HA was incubated with 0.2 µg anti-HA monoclonal antibody (Roche, cat. # 11867423001, clone 3F10) and pre-equilibrated DynaBeads Protein G (Life Technologies) (5 µL bed volume) in a 100 µL reaction at 8°C for three hours. The beads were then washed three times with the HRB-Na100 buffer to remove unbound Ded1p-HA protein, and subsequently mixed with 300 nM S•Tag-eIF4G, 1500 nM eIF4A, 20 µg/mL RNases A,B (Roche), as indicated, in a volume of 100 µL. 3 µL of the reaction volume (including beads) was removed and used as input. The remainder of the reaction volume was incubated under rotation overnight at 8°C. Beads were washed with the HRB-Na100 buffer for three times. Proteins were eluted with Laemmli buffer (Biorad), and denatured at 95°C for 3 min. Proteins were resolved on NuPAGENovex 4–12% Bis-Tris protein gels (1.0 mm) in MOPS buffer (pH 7.7, Life Technology), and visualized by Western blot with primary antibodies against Ded1p (α-Ded1p, polyclonal antibody, raised against full length Ded1p, dilution 1:6000), monoclonal anti-S-Tag antibody, and anti-eIF4A polyclonal antibody (Santa Cruz, prod. # sc-27227).

For the pull-down assays with GST-eIF4A or GST immobilized, 300 µL of the protein-binding reaction was set up with 1500 nM GST-eIF4A or GST, 1500 nM Ded1p$^{117-604}$ or Ded1p$^{1-535}$, 20 µg/mL RNases A,B (Roche), as indicated. 30 µL of the protein-binding reaction was removed as input. The reminder was incubated with rotation overnight at 8°C in the presence of 5 µg/µL BSA (Roche), and was further mixed and incubated with the pre-equilibrated 40 µL (bed volume) glutathione sepharose 4B resin(GE healthcare) at 8°C for one hour. The beads were washed four times with the RIPA buffer. Proteins were eluted with Laemmli Sample Buffer (Biorad), and denatured at 95°C for 3 min. Proteins were resolved on the NuPAGE Novex 4–12% Bis-Tris mini protein gels (1.0 mm thick) in MOPS running buffer (pH7.7) (Life Technology), and visualized either by Western blot with primary antibodies against Ded1p (α-Ded1p, polyclonal antibody, raised against full length Ded1p, dilution 1:6000), or by Coomassie Blue staining for the GST-tagged proteins.

## RNA duplex unwinding

Unwinding reactions were performed under a pre-steady-state regime as previously described (*Yang and Jankowsky, 2005*). Briefly, Ded1p was incubated with 0.5 nM radiolabeled RNA for 5 min in reaction buffer, and reactions were initiated with ATP/Mg$^{2+}$. Where applicable, eIF4G, eIF4A, or both were included. Aliquots were removed at indicated times, and reactions were stopped and applied to 15% nondenaturing PAGE. Gels were dried and RNAs visualized and quantified with a Molecular Dynamics Phosphorimager and the ImageQuant 5.2 software (Molecular Dynamics). Observed rate constants were calculated as described, considering simultaneously occurring strand annealing (*Yang and Jankowsky, 2005*).

## Protein-protein crosslinking

In vitro protein-protein crosslinking with formaldehyde was essentially performed as described (*Putnam et al., 2015*), with Ded1p (0.3 µM), eIF4A (2 µM for *Figure 3C*, or indicated concentrations in *Figure 3—figure supplement 1B*), RNA (2 µM, 16bp, 25 nt 3'-overhang) and 0.5 mM ADPNP-MgCl$_2$ in buffer used in unwinding reactions. Formaldehyde [1% (v/v)] was added, and incubated at room temperature for 30 min. Crosslinking was quenched with 0.5 mM Tris-Glycine (pH 6.8). Proteins were applied to and resolved on the NuPAGENovex 4–12% Bis-Tris protein gels (1.5 mm) in MOPS buffer (pH7.7, Life Technology). Proteins were visualized by Western blot with polyclonal antibody against Ded1p.

## Microscale thermophoresis

EIF4A was labeled with Alexa Fluor 647 NHS ester (ThermoFisher Scientific # A37573) according to the supplied protocol. Excess dye was removed by buffer exchange into 50 mM Tris-HCl (pH 8.0), 2 mM DTT, 1 mM EDTA, 0.1% Triton-X 100, 30% glycerol and 100 mM NaCl using *Zeba*spin desalting columns (0.5 mL, 7K MWCO, ThermoFisher Scientific # 89882). The protein concentration was determined by SDS-PAGE and Coomassie Blue staining, with the diluted BSA (Roche) and unlabeled eIF4A in known concentration as standards.

Microscale thermophoresis measurements were performed at room temperature on a Monolith NT.115 system (NanoTemper Technologies) with standard treated glass capillaries (NanoTemper Technologies). Buffers were identical to those used in unwinding reactions, complemented with 0.5 µg/µL BSA (Roche) and 0.05% (v/v) Tween, to prevent absorption of the proteins to the glass capillaries, and with 25 µg/mL RNase A (USB) to remove residual RNA. 20% IR-laser power was used to locally generate the temperature jump in the capillaries. Fluorescence change in the heated field was monitored with 20% LED power. Laser on and off times were set at 30 s and 5 s, respectively. The eIF4A-eIF4G complex was formed for at least 10 min prior to the addition of Ded1p. Measurements were performed multiple times, averages of individual points were calculated and the resulting binding isotherm was fitted with the quadratic binding equation.

## Thermodynamic model and data fit

Kinetic modeling was performed with KINETK EXPLORER (Kintek, Austin, TX) (*Johnson et al., 2009a*, *2009b*) using data obtained in this and earlier studies (*Putnam et al., 2015*). The thermodynamic framework to describe the interactions between Ded1p, eIF4A, eIF4G, RNA, and ATP was built on the previously established kinetic framework for duplex unwinding by the Ded1p oligomer (*Putnam et al., 2015*). eIF4A and eIF4G only interact with a Ded1p monomer (*Figure 3*). The resulting framework contains 108 binding reactions, listed in *Supplementary file 2*. The model was constrained by considering energy conservation. That is, the change in free energy for transition of one complex into another is independent of the reaction pathway. In addition, we incorporated the experimentally determined RNA affinity of eIF4G (*Putnam et al., 2015*). We further considered that the comparably weak RNA independent ATPase activity of Ded1p was not affected by eIF4G and eIF4A to a measurable degree (data not shown). This notion allowed us to use the Ded1p affinity for ATP in the absence of RNA for multiple Ded1p complexes. Collectively, these considerations enabled us to describe the entire framework in 16 groups of thermodynamically linked binding steps (*Supplementary file 3*).

Ten complexes are theoretically capable of unwinding: Ded1p, eIF4A, eIF4-eIF4G, Ded1p-eIF4G, Ded1p-eIF4A (with ATP bound to Ded1p, eIF4A, or both), Ded1p-eIF4A-eIF4G (with ATP bound to Ded1p, eIF4A, or both). Given that eIF4A$^{E171A}$ (ATPase deficient) was able to stimulate unwinding by Ded1p, while Ded1p$^{E307A}$ (ATPase deficient) did not stimulate unwinding by eIF4A (*Figure 2*), we considered ATP binding to eIF4A irrelevant for the unwinding activity of the Ded1p-eIF4A complex. In the presence of eIF4G, ATP binding by eIF4A was required for the impact on Ded1p (*Figure 2*). We therefore considered the Ded1p-eIF4A-eIF4G complex with ATP bound to both helicases as active unwinding complex. Collectively, these considerations simplified the number of unwinding complexes to six: Ded1p, eIF4A, eIF4-eIF4G, Ded1p-eIF4G, Ded1p-eIF4A (with ATP bound to Ded1p independent of ATP binding to eIF4A), Ded1p-eIF4A-eIF4G (with ATP bound to Ded1p and eIF4A). We did not include ATP hydrolysis steps in our analysis for any complex, except for Ded1p alone, for which we had previously characterized the respective steps (*Putnam et al., 2015*).

Global data fitting was performed using Kintek Global Explorer. 20 datasets for unwinding of the 16 bp duplex with 3'-25 ntssRNA overhang were fit simultaneously. A dataset includes all time courses for titration of 1 component (e.g., protein or ATP) with other components held constant. For each complex at least one dataset was included for a titration of each component (e.g., Ded1-eIF4G: 1 Ded1p titration, 1 eIF4G titration, 1 ATP titration).Datasets were as follows: Ded1p-eIF4A (4 datasets), Ded1p-eIF4G (3 datasets), eIF4A-eIF4G (4 data sets), and Ded1p-eIF4A-eIF4G (9 datasets). Multiple iterations were performed until the best fit to all data sets was achieved. Data were fit assuming that all binding steps for Ded1p with eIF4A, and eIF4G were in rapid equilibrium. Parameters for the best fit values are reported in *Supplementary file 2*. Variables not well constrained by the data are reported as limits. Errors are not always symmetrically distributed around the best fit value (*Figure 4—figure supplement 5*). Uncertainty in each parameter is presented as a range calculated at the 95% confidence interval. To visually assess the quality of fit, experimental data were plotted versus values predicted with the model (*Figure 4B*).

## Acknowledgements

We thank Dr. Jerry Pelletier (McGill Univ., Montreal) for Hippuristanol, Dr. Derek Taylor (Case Western Reserve University, Cleveland) for SUMO plasmids, Dr. Michael Altmann (Univ. Bern, Berne) for the plasmid containing wild-type His-tagged eIF4A, Dr. Jeff Coller (Case Western Reserve University, Cleveland) for yeast strains, Dr. Yinghua Chen and the Protein Expression Purification Crystallization and Molecular Biophysics Core (Case Western Reserve University, Cleveland) for assistance with MST measurements, Sukanya Srinivasan (Case Western Reserve University, Cleveland) for assistance with protein purification, and Drs. Michael Harris and William Merrick (Case Western Reserve University, Cleveland) for helpful discussions.

## Additional information

### Funding

| Funder | Grant reference number | Author |
|---|---|---|
| National Institute of General Medical Sciences | GM118088 | Zhaofeng Gao<br>Andrea A Putnam<br>Heath A Bowers<br>Ulf-Peter Guenther<br>Xuan Ye<br>Eckhard Jankowsky |

The funders had no role in study design, data collection and interpretation, or the decision to submit the work for publication.

### Author contributions

ZG, AAP, Conception and design, Acquisition of data, Analysis and interpretation of data, Drafting or revising the article; HAB, Conception and design, Contributed unpublished essential data or reagents; U-PG, Conception and design, Acquisition of data, Analysis and interpretation of data; XY, AK, Acquisition of data, Analysis and interpretation of data; AKH, Acquisition of data, Analysis and interpretation of data, Drafting or revising the article; EJ, Conception and design, Analysis and interpretation of data, Drafting or revising the article

### Author ORCIDs

Xuan Ye, http://orcid.org/0000-0002-3157-7537
Eckhard Jankowsky, http://orcid.org/0000-0001-7677-7412

## Additional files

### Supplementary files

• Supplementary file 1. RNA oligonucleotides ,primers, plasmids and yeast strains used in this study.

• Supplementary file 2. Parameters calculated from the thermodynamic model.

• Supplementary file 3. Thermodynamically linked binding steps used for modeling.

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
