## [Decision Letter]

Thank you for submitting your article "Coupling between the DEAD-box RNA helicases Ded1p and eIF4A" for consideration by *eLife*. Your article has been reviewed by four peer reviewers, one of whom is a member of our Board of Reviewing Editors, and the evaluation has been overseen James Manley as the Senior Editor. The reviewers have opted to remain anonymous.

The reviewers have discussed the reviews with one another and the Reviewing Editor has drafted this decision to help you prepare a revised submission.

Summary of work:

This paper presents the novel findings that eIF4A, alone and together with eIF4G, allosterically modulates the rate and substrate preferences of RNA unwinding by helicase Ded1 in the context of complexes containing Ded1 and eIF4A, or all three proteins. They do an extensive set of unwinding assays with Ded1 and these other two proteins over a range of different protein, RNA, and ATP concentrations, allowing them to model a thermodynamic framework that generates functional affinity constants (*K_1/2_*values) for forming all of the different potential combinations of these 3 proteins with and without RNA and ATP. Based on these affinity constants and known information about cellular concentrations, they argue that the trimeric complex of Ded1/eIF4A/eIF4G is the predominant form in cells. This would change the way we think about Ded1 as functioning independently of eIF4F (4E/4G/4A) on a subset of mRNAs with highly structured 5'UTRs towards the view that it generally functions as a subunit within eIF4F and that its activity is modulated by eIF4F components. They also have evidence that eIF4A and eIF4G activities are likewise modulated by Ded1. In addition, they examine the effects of eIF4A and eIF4G on unwinding by Ded1 for substrates with 3' versus 5' overhangs, finding the greater stimulation for the former. And they measure unwinding activities for substrates of different duplex lengths from which they deduce different effects of eIF4A alone or together with eIF4G on unwinding by the trimeric complex.

After consultation with the reviewers, the Reviewing Editor came to the decision that it is necessary for you to address all of the essential comments provided in all four reviews (shown below), except for the suggestion of Reviewer #2 that you determine unwinding rates in yeast cytoplasmic extracts that have eIF4A and Ded1p mutants; and the suggestion of Reviewer #3 that you employ Surface Plasmon Resonance to measure the on- and off-rates of the heterotrimeric eIF4G/eIF4A/Ded1 complex. It is also requested that you make the necessary revisions to address each of the less critical comments of the reviewers, as you see fit.

Reviewer #1:

The paper by Gao et al. presents a tour-de-force kinetic analysis of the interactions between Ded1p, eIF4A, eIF4G, RNA and ATP during ongoing unwinding reaction. The data analysis is carried out by global modeling of a huge dataset using KinTek Explorer. The datasets are comprehensive and more than sufficient to make the modeling. In addition, key statements are tested directly by alternative methods which further strengthen the conclusions. This type of analysis is extremely challenging and innovative. The results provide a new insight into the function of helicases in eukaryotic translation and demonstrate the importance of the protein networks in shaping their functional activity. I have only one uncertainty concerning the statistical evaluation of data. In my opinion, correlations between the observed and calculated constants are only the first step in the assessment of the data fit and alone it is insufficient to judge the quality of the fit. Another key piece of fitting is the error analysis for the lower and upper bounds of the calculated values. This is not documented sufficiently. It would be important to provide the information about the σ value, Ch2 Threshold and provide a supplemental figure with the symmetry distributions (see of KinTek manual).

Reviewer #2:

This study proposes that Ded1p is the primary unwinding helicase of the preinitiation complex and that eIF4A plays a rather minimal role in restructuring mRNA. If true, this would be a very important finding and transform our understanding of initiation in yeast. Nevertheless, while the in vitro data are convincing for the complexes studied, one does wonder if the in vitro reconstituted complex truly reflects what occurs in vivo and whether eIF4A unwinding activity is increased by an untested accessory factor (e.g. eIF4B in the presence of Dedp1). One possible set of experiments to test this would be to determine unwinding rates in yeast cytoplasmic extracts that have eIF4A and Ded1p mutants.

It is certainly possible that the function of eIF4G and eIF4B in promoting eIF4A unwinding activity is not conserved between yeast and humans. This raises a rather important point, which is that there now appears to be a rather significant difference between yeast and human eIF4A unwinding potential using purified components in vitro. Is it that the human eIF4G and eIF4B provide the necessary regulation of eIF4A activity, which the yeast homologs just can't accomplish (they are certainly quite different at the sequence level)? Or, is yeast eIF4A naturally a poor unwinding enzyme even in the presence of eIF4G and eIF4B and therefore requires Ded1p? The authors write the introduction as if the two systems can be viewed as the same mechanism, but this is increasingly becoming less likely. As it stands, the authors should rewrite their introduction to raise this fact. While I appreciate stating the link between eIF4A and disease (cancer), the function of yeast eIF4A may have nothing to do with how the human system is regulated.

Some specific points:

The authors provide pull-down data to test if eIF4G-DED1-eIF4A can form a trimeric complex. A limitation of this approach is that the amount of eIF4A pulled down with eIF4G or ded1 is extremely small. As a result, the interpretation that a trimeric complex is formed from this data is not convincing.

In the absence of RNA, equilibrium dissociation constants are determined by microscale thermophoresis. Interestingly, this assay generates rather different data to that previously published for yeast proteins by the Lorsch lab (Mitchell et al. Mol. Cell. 2010 Sep 24;39(6):950-62; not cited). In the Lorsch study, eIF4A binds to an eIF4G-eIF4E complex with an equilibrium dissociation constant that is calculated to be <30 nM. In the current study the apparent affinity of eIF4A to an eIF4G-eIF4E complex (the authors state that eIF4E is in all eIF4G complexes on in the Results section) with an apparent affinity of 500 nM. An order of magnitude difference in measured affinity seems rather large. Do the authors have any explanation for this? These kinds of discrepancies could seriously alter the proposed models of complex formation between components.

An equilibrium dissociation constant of Ded1p to an eIF4G-eIF4A complex is measured to be 7 nM. This doesn't seem to correspond to such a low amount of this complex obtained in the pull-down assay (In the absence of RNA and ATP; Figure 1). Wouldn't one expect a higher amount of protein retained in the pull-down assay with what must be a relatively slow off-rate (in the absence of ATP and RNA)? The explanation that this complex may not be long lived in the presence of ATP and RNA does not explain why these proteins are not retained on the pull-down in the absence of ATP and RNA. This could be tested by showing that the interaction of an ATPase dead mutant of eIF4A and Ded1p to eIF4G should be enhanced in cells.

It is briefly mentioned that eIF4E is also present in the unwinding assays to preserve eIF4G stability. Human eIF4E has been shown to promote duplex unwinding by eIF4A – have the authors tested this mechanism, or is this also a human specific function of eIF4E?

Human eIF4A has recently been shown to possess a very precise step size no matter what accessory protein(s) are bound to it – this required single molecule data to definitely prove (Science. 2015 Jun 26;348(6242):1486-8; not cited). Do the authors really feel that step size is going to change for yeast factors in the presence of accessory proteins compared with human factors? Since the bulk assays provided here can't monitor processivity this claim seems unsupported.

*Reviewer #3:*

This paper presents the novel findings that eIF4A, alone and together with eIF4G, allosterically modulates the rate and substrate preferences of RNA unwinding by helicase Ded1 in the context of complexes containing Ded1 and eIF4A, or all three proteins. They do an extensive set of unwinding assays with Ded1 and these other two proteins over a range of different protein, RNA, and ATP concentrations, allowing them to model a thermodynamic framework that generates functional affinity constants (*K_1/2_*values) for forming all of the different potential combinations of these 3 proteins with and without RNA and ATP. Based on these affinity constants and known information about cellular concentrations, they argue that the trimeric complex of Ded1/eIF4A/eIF4G is the predominant form in cells. This would change the way we think about Ded1 as functioning independently of eIF4F (4E/4G/4A) on a subset of mRNAs with highly structured 5'UTRs towards the view that it generally functions as a subunit within eIF4F and that its activity is modulated by eIF4F components. They also have evidence that eIF4A and eIF4G activities are likewise modulated by Ded1. In addition, they examine the effects of eIF4A and eIF4G on unwinding by Ded1 for substrates with 3' versus 5' overhangs, finding the greater stimulation for the former. And they measure unwinding activities for substrates of different duplex lengths from which they deduce different effects of eIF4A alone or together with eIF4G on unwinding by the trimeric complex.

Critical comments:

Figure 1 provides evidence for Ded1-eIF4A interaction, and shows that addition of eIF4A increases the binding of eIF4G to Ded1, but it's very puzzling that less eIF4A binds to Ded1 in the presence of eIF4G than in its absence, directly at odds with later measurements of Kd values from the thermodynamic modeling – as if interaction between eIF4A and 4G competes with Ded1-eIF4A interaction. In Figure 1, the increases in Ded1 and eIF4A interaction with eIF4G when all three components are present are quite modest, perhaps ~3-fold or less, whereas the thermodynamic modeling later in the paper predicts a 3-order of magnitude increase in affinity for the heterotrimeric complex compared to all 3 dimeric complexes. The authors suggest later in the paper that the Ded1 interaction with eIF4G/eIF4A will have a high off-rate owing to ATP hydrolysis; however, the affinities predicted by the thermodynamic model for the heterotrimeric complex are subnanomolar even in the absence of ATP and RNA-the conditions of the pull-down experiments. One wonders whether the on-rate would have to exceed the diffusion limit to account for the pull-down data on the hetertrimeric complex. Later on, in Figure 5, they measure the affinity of Ded1 for a pre-formed eIF4G-eIF4A complex in vitro with microscale thermophoresis as <7nM (the limit of the instrument). This is consistent with the high affinity deduced from the thermodynamic model; however, it would be superior to employ Surface Plasmon Resonance to confirm that the heterotrimeric complex has a high off-rate, albeit much slower than the on-rate, in the manner required to unify their pull-down data in Figure 1 (and later in Figure 3—figure supplement 2A) with the other measurements of the Kd for the heterotrimeric complex. The important discrepancy between the different approaches used to establish the stability of the key heterotrimeric complex needs to be resolved.

In addition, the results in Figure 1 need to quantified and expressed as the averages of% input bound, calculated from replicate binding experiments.

It's interesting in Figure 2 that an inactive eIF4A mutant can stimulate Ded1 unwinding; which is reversed by eIF4G, supporting their claim that Ded1 is the unwinding activity in Ded1/eIF4A complexes, but the increase in activity over Ded1 alone seen for the Ded1/eIF4A-mut combination is less than that observed with WT eIF4A. Presumably, they want to propose that eIF4A is allosterically stimulating Ded1 activity, and ATP binding to eIF4A enhances this effect without being required for it. But don't these results also argue that Ded1 is not the sole unwinding activity in the complex and that eIF4A also makes an important contribution to it, with Ded1 enhancing eIF4A activity? This latter possibility is also suggested by the findings in Figure 2 that the inhibitory effect of hippurstanol on the Ded1-eIF4A complex is much greater than on eIF4A alone.

Figure 3 shows that 4A can stimulate by ~5x the unwinding activity of a C-terminally truncated Ded1, but not that of an N-terminally truncated Ded1 protein. 4G has no effect on the former, which makes sense, but stimulates the latter even though 4G inhibits WT Ded1. They give an ad hoc explanation for these results about eIF4G increasing RNA binding, but it doesn't make sense because they acknowledge that eIF4G increases RNA binding by both WT Ded1 and the N-terminally truncated Ded1, yet has opposite effects on unwinding by these two Ded1 proteins.

In Figure 3—figure supplement 2A, the pulldown of C-term truncated Ded1 with GST-eIF4A is barely observed, which is the positive control for the failure of the N-term truncated Ded1 to be pulled down with GST-eIF4A, again indicating an extremely weak interaction between these proteins in vitro. As above, quantification of replicate experiments is required to substantiate these results.

It should be examined whether overexpression of eIF4A does not suppress the cold-sensitive phenotype of the N-term truncation of Ded1. If so, this would provide needed evidence that the genetic suppression observed in Figure 1 for ded1-TAM is dependent on Ded1-eIF4A interaction and thus a manifestation of the ability of eIF4A to enhance Ded1 function rather than acting in parallel to Ded1.

Figure 3 is not entirely convincing. It would improve the analysis to show by Western blotting that eIF4A is present in the deduced Ded1-eIF4A heterodimer, and that addition of additional eIF4A can eliminate all of the Ded1-Ded1 dimer and increase the amount of the Ded1-eIF4A heterodimer.

Reviewer #4:

I recommend acceptance after minor edits.

This manuscript investigates functional and physical interactions of the yeast DEAD-box protein Ded1p with eIF4F complex components eIF4A and eIF4G. Using quantitative biochemical approaches the authors show that Ded1 and eIF4A interact with one another and simultaneously interact with the scaffold protein eIF4G. Utilizing a framework of binding and unwinding reactions with different protein combinations, they characterize thermodynamic stability of all three possible binary complexes as well as the ternary complex (eIF4A-eIF4G-Ded1p), and find that the ternary complex is the most stable species. They find that while all complexes are competent for unwinding, the ternary complex exhibits the highest unwinding activity. Importantly, the authors show that within the ternary complex, Ded1 is responsible for the bulk of ATP-dependent RNA unwinding/remodeling activity, while eIF4A and eIF4G are solely stimulatory. The authors suggest that Ded1 is an integral and dynamically associating component of eIF4F complex. In addition, they identify the N-terminus of Ded1p as the region responsible for its interaction with eIF4A. Combined with their previous finding that the C-terminus of Ded1 makes interaction with eIF4G, this allows the authors to hypothesize about the overall structural organization of the ternary complex.

The paper establishes a previously missing functional and physical link between the two key DEAD-box proteins involved in translation initiation, Ded1 and eIF4A. It presents compelling evidence that Ded1p may be regarded as a dynamically associating component of eIF4F complex. The latter is an important and potentially paradigm-shifting finding for the field of translation regulation as well as our understanding of the early events in translation initiation. The conventional view of translation initiation suggests that eIF4F is comprised of well-defined and stably associated components, and the same eIF4F complex acts on all mRNAs. The new evidence presented here shows that dynamic association of Ded1p with eIF4A and eIF4G can result in an array of complexes with variable properties, which paints a much more dynamic picture of early initiation events. Jankowsky and coworkers also show that, depending on their structure (i.e., 3'-tailed vs 5'-tailed duplex), RNA substrates bind different complexes with different affinities. This suggests that populations of mRNAs with different 5' UTR structures can be differentially expressed depending on the profile of prevalent eIF4F complexes. Importantly, they show that even relatively small changes in expression levels (~2-fold) of Ded1 or eIF4G can lead to drastic changes in the abundance of the binary and ternary complexes, and consequently to large effects on translation initiation efficiency.

This is a comprehensive study with well-designed experiments and thorough data analysis that is sure to be of interest to researchers in the field of translation regulation as well as the general readership of *eLife*. Most figures are clear and informative and the paper is written in a concise and organized language. Nonetheless, a few points still require clarification.

1) The paper would benefit from added discussion on how the new data can be interpreted in the light of earlier findings from Jankowsky and Parker lab (Hilliker, 2011) on the role of Ded1 as both translation activator and repressor. In my opinion, this should be an important part of the Discussion, and currently there is almost no mention of it.

2) The thermodynamic framework based on fitting multiple unwinding data sets is an important part of the paper as it allows calculating the abundance of different complexes based on concentrations of their components. However, the paper provides very little detail on how this analysis was carried out. This makes it difficult to evaluate the robustness of their analysis. The authors need to describe this in much more detail.

3) The representation of the framework of interactions in Figure 4 is very confusing. I suggest the conventional way of showing reactions with double arrows, and making this panel one of the supplementary figures.

4) Using pie charts to show complex composition shown in Figure 7 doesn't seem ideal. It is confusing and contains redundant information. Is there any better way to present this? Perhaps a ternary plot?

Finally, a couple of minor things:

Figure 4 and corresponding legend: "A" and "T" are used interchangeably to refer to ATP.

Figure 4: What is this unwinding rate, exactly? The legend doesn't say.

---

## [Author Response]

*Reviewer #1:*

*The paper by Gao et al. presents a tour-de-force kinetic analysis of the interactions between Ded1p, eIF4A, eIF4G, RNA and ATP during ongoing unwinding reaction. The data analysis is carried out by global modeling of a huge dataset using KinTek Explorer. The datasets are comprehensive and more than sufficient to make the modeling. In addition, key statements are tested directly by alternative methods which further strengthen the conclusions. This type of analysis is extremely challenging and innovative. The results provide a new insight into the function of helicases in eukaryotic translation and demonstrate the importance of the protein networks in shaping their functional activity.*

*I have only one uncertainty concerning the statistical evaluation of data. In my opinion, correlations between the observed and calculated constants are only the first step in the assessment of the data fit and alone it is insufficient to judge the quality of the fit. Another key piece of fitting is the error analysis for the lower and upper bounds of the calculated values. This is not documented sufficiently. It would be important to provide the information about the σ value, Ch2 Threshold and provide a supplemental figure with the symmetry distributions (see page 66 of KinTek manual).*

We have now included a supplementary figure (Figure 4—figure supplement 5) to illustrate Χ^2^ symmetry distributions for two cases, one where increase and decrease in the parameter (*K_1/2_*) relative to the optimal value results in a poorer fit, and another where only a decrease of the parameter, but not an increase result in a poorer fit. We trust this figure sufficiently explains the 95% confidence intervals that we have listed for each parameter in the [Supplementary-material SD2-data]. The symmetry distributions for each Χ^2^ value are apparent in these tables from the respective lower and upper limit for each parameter relative to the optimal parameter.

For the visual assessment of the data fit, we feel that displaying the correlation between calculated and experimental values for the observed unwinding rate constants is the most intuitive way to visually inspect the quality of the datafit, along with respective plots for subgroups of data (Figure 4—figure supplement 6). We favor this representation over plots for σ values for individual datasets because of the sheer number of the datasets used. Plots for σ values for individual datasets would, in our opinion, render a visual inspection of the model quality essentially impossible. Our plot, in contrast, allows a global assessment of the model quality

*Reviewer #2:*

*This study proposes that Ded1p is the primary unwinding helicase of the preinitiation complex and that eIF4A plays a rather minimal role in restructuring mRNA. If true, this would be a very important finding and transform our understanding of initiation in yeast. Nevertheless, while the in vitro data are convincing for the complexes studied, one does wonder if the in vitro reconstituted complex truly reflects what occurs in vivo and whether eIF4A unwinding activity is increased by an untested accessory factor (e.g. eIF4B in the presence of Dedp1). One possible set of experiments to test this would be to determine unwinding rates in yeast cytoplasmic extracts that have eIF4A and Ded1p mutants.*

*It is certainly possible that the function of eIF4G and eIF4B in promoting eIF4A unwinding activity is not conserved between yeast and humans. This raises a rather important point, which is that there now appears to be a rather significant difference between yeast and human eIF4A unwinding potential using purified components in vitro. Is it that the human eIF4G and eIF4B provide the necessary regulation of eIF4A activity, which the yeast homologs just can't accomplish (they are certainly quite different at the sequence level)? Or, is yeast eIF4A naturally a poor unwinding enzyme even in the presence of eIF4G and eIF4B and therefore requires Ded1p? The authors write the introduction as if the two systems can be viewed as the same mechanism, but this is increasingly becoming less likely. As it stands, the authors should rewrite their introduction to raise this fact. While I appreciate stating the link between eIF4A and disease (cancer), the function of yeast eIF4A may have nothing to do with how the human system is regulated.*

We have removed the reference to the link between eIF4A and cancer. We have also carefully examined the remainder of the introduction for statements that could be understood as implicit or explicit suggestion that yeast and human eIF4A-eIF4G and associated factors could be viewed as using the same mechanism. In the current form, the introduction does not contain any such statements.

Since our work does not test the mammalian system we have been careful to avoid any suggestion to similarities or differences between mammalian and yeast translation initiation machineries, other than stating the undeniable sequence similarity between yeast and mammalian eIF4A, and between yeast Ded1p and mammalian DDX3.

*Some specific points:*

*The authors provide pull-down data to test if eIF4G-DED1-eIF4A can form a trimeric complex. A limitation of this approach is that the amount of eIF4A pulled down with eIF4G or ded1 is extremely small. As a result, the interpretation that a trimeric complex is formed from this data is not convincing.*

We agree with the reviewer that the pull-down data alone would not adequately permit to conclude the existence of a trimeric complex. For this reason we have stated that the data suggest that all three proteins can interact simultaneously. We feel that this statement adequately reflects the data. Most importantly, however, we provide several orthogonal lines of functional evidence that all three proteins can interact simultaneously.

*In the absence of RNA, equilibrium dissociation constants are determined by microscale thermophoresis. Interestingly, this assay generates rather different data to that previously published for yeast proteins by the Lorsch lab (Mitchell et al. Mol. Cell. 2010 Sep 24;39(6):950-62; not cited). In the Lorsch study, eIF4A binds to an eIF4G-eIF4E complex with an equilibrium dissociation constant that is calculated to be <30 nM. In the current study the apparent affinity of eIF4A to an eIF4G-eIF4E complex (the authors state that eIF4E is in all eIF4G complexes on in the Results section) with an apparent affinity of 500 nM. An order of magnitude difference in measured affinity seems rather large. Do the authors have any explanation for this? These kinds of discrepancies could seriously alter the proposed models of complex formation between components.*

We are aware of the noted study that reported several dissociation rate constants for the eIF4A-eIF4G complex. We apologize for inadvertently omitting the reference, which is now included.

Before specifically addressing the reviewer's point, we emphasize that the values reported in this study, with the exception of the noted dissociation constant for eIF4A binding to the eIF4G-eIF4E complex, are in excellent agreement with our parameters. Given that the values in both studies were measured by fundamentally different approaches, the overall agreement speaks for the robustness of the protein system and for all of the used approaches.

Having said this, we do appreciate the difference between the dissociation constants for eIF4A binding to the eIF4G-eIF4E complex, which are an order of magnitude larger in our measurements than in the previous study. Given the agreement of the other values in both studies, we believe that the difference reflects a distinction between our functional approach (based on enzymatic activity) vs. the equilibrium binding used in the previous study. In the fluorescence-based equilibrium binding, proximity of eIF4A to eIF4G-eIF4E is measured. For our measurements, the complex resulting in eIF4A binding to eIF4G-eIF4E only shows an impact if it further interacts with Ded1p and thereby affects Ded1p's unwinding activity. It is possible that eIF4A can bind to eIF4G-eIF4E to form a complex that does not interact with Ded1p, and that this complex forms with higher affinity than seen in our data. If the binding and dissociation dynamics of this complex are fast compared to the timescale of our experiments, our model would be insensitive to such a complex. As with all models for reaction systems that contain multiple components, our framework contains only the minimally required stages to describe the observed data. Additional physical stages for components that do not directly participate in the reaction can never be ruled out. Thus, the noted discrepancy does not alter the proposed models of complex formation between components.

*An equilibrium dissociation constant of Ded1p to an eIF4G-eIF4A complex is measured to be 7 nM. This doesn't seem to correspond to such a low amount of this complex obtained in the pull-down assay (In the absence of RNA and ATP; Figure 1). Wouldn't one expect a higher amount of protein retained in the pull-down assay with what must be a relatively slow off-rate (in the absence of ATP and RNA)? The explanation that this complex may not be long lived in the presence of ATP and RNA does not explain why these proteins are not retained on the pull-down in the absence of ATP and RNA. This could be tested by showing that the interaction of an ATPase dead mutant of eIF4A and Ded1p to eIF4G should be enhanced in cells.*

We agree that the amounts of proteins retained in the pull-down experiments appear to not very well correspond to the dissociation constants determined later. However, pull-down assays are by their very nature not conducted under equilibrium conditions. As described in the Methods section, our pull-down assays involve several stringent washings steps, which had to be included to provide a clear signal to background ratio. Pull-down assays that include Ded1p had to be particularly stringently washed, to remove Ded1p that non-specifically adhered to the beads. We speculate that the high level of non-specific adherence is seen because Ded1p contains on both N- and C-termini regions of low complexity that are also very basic. In addition, surface immobilization of a component does affect binding behavior, by (a) affecting the degrees of freedom of the system, and (b) creating a microenvironment that might be unfavorable to binding. Given the considerations above, it is perhaps not surprising that the amount of complex retained on beads does not closely correspond to the apparent dissociation constants measured in solution. It is well established that pull-down experiments often reflect solution affinities poorly, especially for dynamic systems. We nevertheless decided to include the pull-down data to provide a qualitative visualization of the protein-protein interactions. However, we emphasize that we quantitatively examine the interactions with suitable functional solution methods.

With regard to the suggested experiment examining whether the interaction of an ATPase-dead mutant of eIF4A and Ded1p to eIF4G is enhanced in cells, we note that Ded1p is an essential protein and that a variant with inactivated ATPase activity is dominant negative in vivo (Hilliker et al., 2011). For this reason, the proposed experiment is not technically feasible.

*It is briefly mentioned that eIF4E is also present in the unwinding assays to preserve eIF4G stability. Human eIF4E has been shown to promote duplex unwinding by eIF4A – have the authors tested this mechanism, or is this also a human specific function of eIF4E?*

We have not explicitly tested the impact of eIF4E alone of eIF4A. However, we have examined the impact of eIF4G-eIF4E on eIF4A (Figure 1, Figure 2). It is formally possible that the impact on eIF4A is in part due to eIF4E.

*Human eIF4A has recently been shown to possess a very precise step size no matter what accessory protein(s) are bound to it – this required single molecule data to definitely prove (Science. 2015 Jun 26;348(6242):1486-8; not cited). Do the authors really feel that step size is going to change for yeast factors in the presence of accessory proteins compared with human factors? Since the bulk assays provided here can't monitor processivity this claim seems unsupported.*

The reviewer refers to a study where processive duplex unwinding by mammalian eIF4A is reported in an optical tweezers setting. Ded1p does not make consecutive unwinding steps, and therefore step size or processivity are not instructive parameters. The strand separation step we are referring to describes the strand opening event by Ded1p, as defined in Putnam et al. (2015) and Yang et al., (2007). The data in Figure 6 show that eIF4A and eIF4G-eIF4A impact the number of basepairs opened by Ded1p in this strand separation step, the time the opened strands are held apart, or both. The important take-home message from these data is that eIF4A and eIF4G-eIF4A affect the strand opening step by Ded1p in a subtle, yet notable fashion. This is what we state.

*Reviewer #3:*

*This paper presents the novel findings that eIF4A, alone and together with eIF4G, allosterically modulates the rate and substrate preferences of RNA unwinding by helicase Ded1 in the context of complexes containing Ded1 and eIF4A, or all three proteins. They do an extensive set of unwinding assays with Ded1 and these other two proteins over a range of different protein, RNA, and ATP concentrations, allowing them to model a thermodynamic framework that generates functional affinity constants (K_1/2_values) for forming all of the different potential combinations of these 3 proteins with and without RNA and ATP. Based on these affinity constants and known information about cellular concentrations, they argue that the trimeric complex of Ded1/eIF4A/eIF4G is the predominant form in cells. This would change the way we think about Ded1 as functioning independently of eIF4F (4E/4G/4A) on a subset of mRNAs with highly structured 5'UTRs towards the view that it generally functions as a subunit within eIF4F and that its activity is modulated by eIF4F components. They also have evidence that eIF4A and eIF4G activities are likewise modulated by Ded1. In addition, they examine the effects of eIF4A and eIF4G on unwinding by Ded1 for substrates with 3' versus 5' overhangs, finding the greater stimulation for the former. And they measure unwinding activities for substrates of different duplex lengths from which they deduce different effects of eIF4A alone or together with eIF4G on unwinding by the trimeric complex.*

*Critical comments:*

*Figure 1 provides evidence for Ded1-eIF4A interaction, and shows that addition of eIF4A increases the binding of eIF4G to Ded1, but it's very puzzling that less eIF4A binds to Ded1 in the presence of eIF4G than in its absence, directly at odds with later measurements of Kd values from the thermodynamic modeling – as if interaction between eIF4A and 4G competes with Ded1-eIF4A interaction. In Figure 1, the increases in Ded1 and eIF4A interaction with eIF4G when all three components are present are quite modest, perhaps ~3-fold or less, whereas the thermodynamic modeling later in the paper predicts a 3-order of magnitude increase in affinity for the heterotrimeric complex compared to all 3 dimeric complexes. The authors suggest later in the paper that the Ded1 interaction with eIF4G/eIF4A will have a high off-rate owing to ATP hydrolysis; however, the affinities predicted by the thermodynamic model for the heterotrimeric complex are subnanomolar even in the absence of ATP and RNA-the conditions of the pull-down experiments. One wonders whether the on-rate would have to exceed the diffusion limit to account for the pull-down data on the hetertrimeric complex. Later on, in Figure 5, they measure the affinity of Ded1 for a pre-formed eIF4G-eIF4A complex in vitro with microscale thermophoresis as <7nM (the limit of the instrument). This is consistent with the high affinity deduced from the thermodynamic model; however, it would be superior to employ Surface Plasmon Resonance to confirm that the heterotrimeric complex has a high off-rate, albeit much slower than the on-rate, in the manner required to unify their pull-down data in Figure 1 (and later in Figure 3—figure supplement 2A) with the other measurements of the Kd for the heterotrimeric complex. The important discrepancy between the different approaches used to establish the stability of the key heterotrimeric complex needs to be resolved.*

As noted in response to a comment by reviewer 2, we agree with the notion that the amounts of proteins retained in the pull-down experiments do not correspond well to the dissociation constants in solution that are determined later in the manuscript. We refer to our response to this comment. We believe the stringent washings steps necessary to achieve a clear pull-down signal emphasize the off rate constants for respective components, which might differ from the trend in dissociation constants. A case in point is perhaps the traditionally weak signal seen in pull down experiments of yeast eIF4A by 4G (or vice versa), even though it is clear that both proteins interact with appreciable affinity. We included the pull-down data to qualitatively visualize protein-protein interactions.

The reviewer makes a keen observation regarding the association rate constants, given the sub-nanomolar dissociation constant and dissociation rate constants that would need to be invoked to account for the low pull-down efficiencies. A back of the envelope calculation using a conservative estimate for the diffusion limited second order association rate constant of k_diff_ ~ 10^9^ M^-1^min^-1^ (the limit is likely higher, which would result in greater dissociation rate constants and shorter half lives) and the obtained dissociation constant of Ded1p binding to eIF4A-eIF4G of *K_1/2_*= 0.7 nM yields a dissociation rate constant of k_off_ ~ 0.7 min^-1^, which corresponds to a half life of the complex of t_1/2_ ~ 1 min. This half life is exceeded many times in the washing steps of the pull down experiments. We therefore believe that observations from the pull-down experiments are consistent with functional data.

We agree with the reviewer in that it would be desirable to measure rate constants for the trimeric complex by a direct method, such as SPR. We have in fact spent a significant amount of time (and money) on SPR experiments to accomplish this goal (before we submitted the manuscript). However, we were unable to identify a suitable chip for use with Ded1p. Inclusion of Ded1p was in all cases associated with a very strong background signal, presumably due to non-specific adherence of Ded1p to the chips. For this reason, we resorted to the use of MST, which is a solution method.

*In addition, the results in Figure 1 need to quantified and expressed as the averages of% input bound, calculated from replicate binding experiments.*

We have included the requested quantitative information in the figure captions.

*It's interesting in Figure 2 that an inactive eIF4A mutant can stimulate Ded1 unwinding; which is reversed by eIF4G, supporting their claim that Ded1 is the unwinding activity in Ded1/eIF4A complexes, but the increase in activity over Ded1 alone seen for the Ded1/eIF4A-mut combination is less than that observed with WT eIF4A. Presumably, they want to propose that eIF4A is allosterically stimulating Ded1 activity, and ATP binding to eIF4A enhances this effect without being required for it. But don't these results also argue that Ded1 is not the sole unwinding activity in the complex and that eIF4A also makes an important contribution to it, with Ded1 enhancing eIF4A activity? This latter possibility is also suggested by the findings in Figure 2 that the inhibitory effect of hippurstanol on the Ded1-eIF4A complex is much greater than on eIF4A alone.*

As the reviewer notes, we are making the point that eIF4A affects the activity of Ded1p. The hippuristanol data also suggest that ATP and RNA binding by eIF4A (without eIF4G) have an impact. This is not surprising, since our observations suggest that eIF4A serves as loading adaptor for a Ded1p protomer, as we note in the discussion. As loading adaptor, eIF4A would bind the unapaired tail of the RNA substrate, and this binding is most likely influenced by the ability of eIF4A to bind ATP, RNA, or both. EIF4A would thus be sensitive to compounds that interfer with either or both of these activities. Yet, the unit responsible for the strand separation is still solely Ded1p. Of course, since Ded1p and eIF4A form a complex, they both ultimately contribute to the activity of the complex.

*Figure 3 shows that 4A can stimulate by ~5x the unwinding activity of a C-terminally truncated Ded1, but not that of an N-terminally truncated Ded1 protein. 4G has no effect on the former, which makes sense, but stimulates the latter even though 4G inhibits WT Ded1. They give an ad hoc explanation for these results about eIF4G increasing RNA binding, but it doesn't make sense because they acknowledge that eIF4G increases RNA binding by both WT Ded1 and the N-terminally truncated Ded1, yet has opposite effects on unwinding by these two Ded1 proteins.*

We apologize for the ad hoc impression of our explanation for the seemingly opposite effects of eIF4G on the N-terminally truncated Ded1p and wtDed1p. The key to appreciate the differences lies in the superposition of several effects – altered RNA affinities and different unwinding activities for the complexes vs. the Ded1p variants without eIF4G. To visualize this point, we have included a schematic figure below, showing observed unwinding rate constants (with and without eIF4G as indicated), as a function of the Ded1p concentration, and the concentration where the reported experiments were conducted.10.7554/eLife.16408.025Author Response Image 1.**DOI:**
http://dx.doi.org/10.7554/eLife.16408.025

*In Figure 3—figure supplement 2A, the pulldown of C-term truncated Ded1 with GST-eIF4A is barely observed, which is the positive control for the failure of the N-term truncated Ded1 to be pulled down with GST-eIF4A, again indicating an extremely weak interaction between these proteins in vitro. As above, quantification of replicate experiments is required to substantiate these results.*

We have included the requested information in the figure caption.

*It should be examined whether overexpression of eIF4A does not suppress the cold-sensitive phenotype of the N-term truncation of Ded1. If so, this would provide needed evidence that the genetic suppression observed in Figure 1 for ded1-TAM is dependent on Ded1-eIF4A interaction and thus a manifestation of the ability of eIF4A to enhance Ded1 function rather than acting in parallel to Ded1.*

We have performed the requested experiments. The data are shown in Figure 3. As the reviewer predicts, overexpression of eIF4A does not suppress the cold-sensitive phenotype of the N-term truncation of Ded1p. We thank the reviewer for the suggesting the experiment, which strengthens the argument for a biologically important interaction between Ded1p and eIF4A.

*Figure 3 is not entirely convincing. It would improve the analysis to show by Western blotting that eIF4A is present in the deduced Ded1-eIF4A heterodimer, and that addition of additional eIF4A can eliminate all of the Ded1-Ded1 dimer and increase the amount of the Ded1-eIF4A heterodimer.*

We have performed the requested crosslinking experiment with varying eIF4A concentration, now shown in Figure 3—figure supplement 1. The data follow the expected trend: the amount of Ded1p dimer decreases with increasing concentrations of eIF4A, and the amount of the Ded1-eIF4A heterodimer increases.

We have also extensively probed the membrane with eIF4A antibody, but the antibody does not detect eIF4A in the complex with Ded1p, presumably because the epitope on eIF4A is occluded in the crosslinked complex. Since we have demonstrated the interaction between Ded1p and eIF4A by several other means, we trust that there is no doubt regarding the eIF4A-Ded1p interaction.

*Reviewer #4:*

*I recommend acceptance after minor edits.*

*This manuscript investigates functional and physical interactions of the yeast DEAD-box protein Ded1p with eIF4F complex components eIF4A and eIF4G. Using quantitative biochemical approaches the authors show that Ded1 and eIF4A interact with one another and simultaneously interact with the scaffold protein eIF4G. Utilizing a framework of binding and unwinding reactions with different protein combinations, they characterize thermodynamic stability of all three possible binary complexes as well as the ternary complex (eIF4A-eIF4G-Ded1p), and find that the ternary complex is the most stable species. They find that while all complexes are competent for unwinding, the ternary complex exhibits the highest unwinding activity. Importantly, the authors show that within the ternary complex, Ded1 is responsible for the bulk of ATP-dependent RNA unwinding/remodeling activity, while eIF4A and eIF4G are solely stimulatory. The authors suggest that Ded1 is an integral and dynamically associating component of eIF4F complex. In addition, they identify the N-terminus of Ded1p as the region responsible for its interaction with eIF4A. Combined with their previous finding that the C-terminus of Ded1 makes interaction with eIF4G, this allows the authors to hypothesize about the overall structural organization of the ternary complex.*

*The paper establishes a previously missing functional and physical link between the two key DEAD-box proteins involved in translation initiation, Ded1 and eIF4A. It presents compelling evidence that Ded1p may be regarded as a dynamically associating component of eIF4F complex. The latter is an important and potentially paradigm-shifting finding for the field of translation regulation as well as our understanding of the early events in translation initiation. The conventional view of translation initiation suggests that eIF4F is comprised of well-defined and stably associated components, and the same eIF4F complex acts on all mRNAs. The new evidence presented here shows that dynamic association of Ded1p with eIF4A and eIF4G can result in an array of complexes with variable properties, which paints a much more dynamic picture of early initiation events. Jankowsky and coworkers also show that, depending on their structure (i.e., 3'-tailed vs 5'-tailed duplex), RNA substrates bind different complexes with different affinities. This suggests that populations of mRNAs with different 5' UTR structures can be differentially expressed depending on the profile of prevalent eIF4F complexes. Importantly, they show that even relatively small changes in expression levels (~2-fold) of Ded1 or eIF4G can lead to drastic changes in the abundance of the binary and ternary complexes, and consequently to large effects on translation initiation efficiency.*

*This is a comprehensive study with well-designed experiments and thorough data analysis that is sure to be of interest to researchers in the field of translation regulation as well as the general readership of eLife. Most figures are clear and informative and the paper is written in a concise and organized language. Nonetheless, a few points still require clarification.*

*1) The paper would benefit from added discussion on how the new data can be interpreted in the light of earlier findings from Jankowsky and Parker lab (Hilliker, 2011) on the role of Ded1 as both translation activator and repressor. In my opinion, this should be an important part of the Discussion, and currently there is almost no mention of it.*

We have very carefully considered possibilities for additional discussion on the role of Ded1 as translation activator and repressor. However, since we have not performed assays that measure specific aspects of translation, our data do not provide much new specific insight into the function of Ded1p as translation activator and repressor, other than implying eIF4A in this function as well. For this reason, and given that the exact physical process of Ded1p function in translation initiation is still elusive, we felt that additional discussion on this point would be too speculative.

*2) The thermodynamic framework based on fitting multiple unwinding data sets is an important part of the paper as it allows calculating the abundance of different complexes based on concentrations of their components. However, the paper provides very little detail on how this analysis was carried out. This makes it difficult to evaluate the robustness of their analysis. The authors need to describe this in much more detail.*

We have included additional table supplements ([Supplementary-material SD3-data]) that list the thermodynamically linked groups of complexes. In addition, as described in response to reviewer #1, we have included a new figure supplement (Figure 4—figure supplement 5) illustrating chi squared symmetry distributions. The Χ^2^ values for each of the more than 100 calculated parameters are given in the [Supplementary-material SD2-data].

*3) The representation of the framework of interactions in Figure 4 is very confusing. I suggest the conventional way of showing reactions with double arrows, and making this panel one of the supplementary figures.*

We appreciate this point. We have tested many different ways to represent the framework, including the suggested double arrows. However, this does make the framework virtually impossible to assess by visual means. We finally settled on the shown representation, because this allowed us to include color bars for dissociation constants. This allows the inclined reader to visually assess differences in dissociation constants for the various complexes. For example, dissociation constants for RNA binding are grouped underneath each other, thus providing what we believe intuitive means to digest the complex data.

We also appreciate the suggestion to move the framework to a supplementary figure. However, since the framework is a pivotal aspect of our paper, we prefer to keep it in a main figure.

*4) Using pie charts to show complex composition shown in Figure 7 doesn't seem ideal. It is confusing and contains redundant information. Is there any better way to present this? Perhaps a ternary plot?*

Again, we appreciate this point. Here, as above, we have extensively experimented with different ways to visualize the data, including ternary plots. We decided against this representation, because we would need to track six distinct species, which would require as many ternary plots. Even then, the important points (presence of Ded1p in the complexes with 4A and 4G at physiological concentrations of the components, and presence of 4G in complexes with Ded1p at physiological concentrations of the components) are not readily apparent, possibly requiring another layer of graphics. We settled on the shown plots, because we felt those showed somewhat intuitively the prevalence of the complexes containing Ded1p as a function of the concentration of 4G and Ded1p.